# MULTI-AGENT POLICY TRANSFER VIA TASK RELATIONSHIP MODELING

## ABSTRACT

Team adaptation to new cooperative tasks is a hallmark of human intelligence, which has yet to be fully realized in learning agents. Previous works on multi-agent transfer learning accommodate teams of different sizes, but heavily rely on the generalization ability of neural networks for adapting to unseen tasks. We posit that the relationship among tasks provides the key information for policy adaptation. To utilize such relationship for efficient transfer, we try to discover and exploit the knowledge among tasks from different teams, propose to learn effect-based task representations as a common latent space among tasks, and use it to build an alternatively fixed training scheme. We demonstrate that the task representation can capture the relationship among teams and generalize to unseen tasks. As a result, the proposed method can help transfer learned cooperation knowledge to new tasks after training on a few source tasks, and the learned transferred policies can also help solve tasks that are hard to learn from scratch.

## 1 INTRODUCTION

Cooperation in human groups is characterized by resiliency to unexpected changes and purposeful adaptation to new tasks (Tjosvold, 1984). This flexibility and transferability of cooperation is a hallmark of human intelligence. Computationally, multi-agent reinforcement learning (MARL) (Zhang et al., 2021a) provides an important means for machines to imitate human cooperation. Although recent MARL research has made prominent progress in many aspects of cooperation, such as policy decentralization (Lowe et al., 2017; Rashid et al., 2018; Wang et al., 2021a;c; Cao et al., 2021), communication (Foerster et al., 2016; Jiang & Lu, 2018), and organization (Jiang et al., 2019; Wang et al., 2020a; 2021b), how to realize the ability of group knowledge transfer is still an open question.

Compared to single-agent knowledge reuse (Zhu et al., 2020), a unique challenge faced by multi-agent transfer learning is the varying size of agent groups. The number of agents and the length of observation inputs in unseen tasks may differ from those in source tasks. To solve this problem, existing multi-agent transfer learning approaches build population-invariant (Long et al., 2019) and input-length-invariant (Wang et al., 2020c) learning structures using graph neural networks (Agarwal et al., 2020) and attention mechanisms like transformers (Hu et al., 2021; Zhou et al., 2021). Although these methods handle varying populations and input lengths well, their knowledge transfer to unseen tasks mainly depends on the inherent generalization ability of neural networks. The relationship among tasks in MARL is not fully exploited for more efficient transfer.

Towards making up for this shortage, we study the discovery and utilization of common structures in multi-agent tasks and propose Multi-Agent Transfer reinforcement learning via modeling TAsk Relationship (MATTAR). In this learning framework, we capture the common structure of tasks by modeling the similarity among transition and reward functions of different tasks. Specifically, we train a forward model for all source tasks to predict the observation, state, and reward at the next timestep given the current observation, state, and actions. The challenge is how to embody the similarity and the difference among tasks in this forward model, we specifically introduce difference by giving each source task a unique representation and model the similarity by generating the parameters of the forward model via a shared hypernetwork, which we call the representation explainer.

To learn a well-formed representation space that encodes task relationship, an alternative-fixed training method is proposed to learn the task representation and representation explainer. During training, representations of source tasks are pre-defined and fixed as mutual orthogonal vectors,

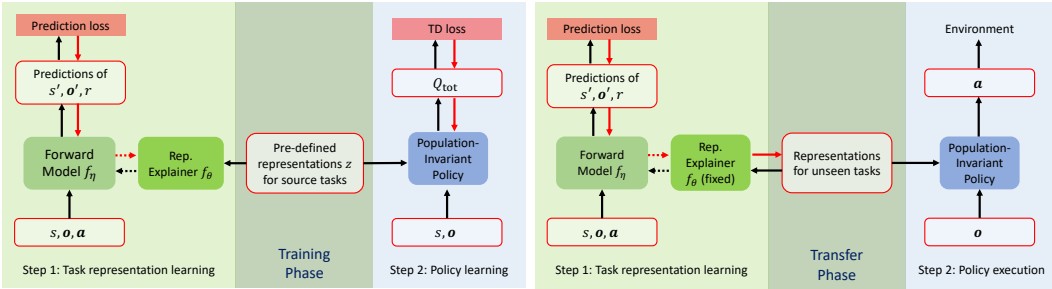

Figure 1: Transfer learning scheme of our method. The black arrows indicate the direction of data flow and the red ones indicate the direction of gradient flow. The dashed arrows indicate the flow between the hypernetwork and the generated network.

and the representation explainer is learned by optimizing the forward model prediction loss on all source tasks. When facing an unseen task, we fix the representation explainer and backpropagate gradients through the fixed forward model to learn the representation of the new task by a few samples. Furthermore, we design a population-invariant policy network conditioned on the learned task representation. During policy training, the representations for all source tasks are fixed, and the policy is updated to maximize the expected return over all source tasks. On an unseen task, we obtain the transferred policy by simply inserting the new task representation into the learned policy network.

On the SMAC (Samvelyan et al., 2019) and MPE (Lowe et al., 2017) benchmarks, we empirically show that the learned knowledge from source tasks can be transferred to a series of unseen tasks with great success rates. We also pinpoint several other advantages brought by our method. First, fine-tuning the transferred policy on unseen tasks achieves better performance than learning from scratch, indicating that the task representation and pre-trained policy network provide a good initialization point. Second, training on multiple source tasks gets better performance compared to training on them individually and other multi-task learning methods, showing that MATTAR also provides a method for multi-agent multi-task learning. Finally, although not designed for this goal, our structure enables comparable performance against single-task learning algorithms when trained on single tasks.

## 2 METHOD

In this paper, we focus on knowledge transfer among fully cooperative multi-agent tasks that can be modeled as a Dec-POMDP (Oliehoek & Amato, 2016) consisting of a tuple $G = \langle I, S, A, P, R, \Omega, O, n, \gamma \rangle$, where $I$ is the finite set of $n$ agents, $s \in S$ is the true state of the environment, and $\gamma \in [0, 1)$ is the discount factor. At each timestep, each agent $i$ receives an observation $o_i \in \Omega$ drawn according to the observation function $O(s, i)$ and selects an action $a_i \in A$. Individual actions form a joint action $\boldsymbol{a} \in A^n$, which leads to a next state $s'$ according to the transition function $P(s'|s, \boldsymbol{a})$, and a reward $r = R(s, \boldsymbol{a})$ shared by all agents. Each agent has local action-observation history $\tau_i \in \mathrm{T} \equiv (\Omega \times A)^* \times \Omega$. Agents learn to collectively maximize the global action-value function $Q_{\text{tot}}(s, \boldsymbol{a}) = \mathbb{E}_{s_{0:\infty}, a_{0:\infty}}[\sum_{t=0}^{\infty} \gamma^t R(s_t, \boldsymbol{a}_t)|s_0 = s, \boldsymbol{a}_0 = \boldsymbol{a}]$ (a little notation abuse: the subscript here for $s$ and $\boldsymbol{a}$ indicates the timestep while the subscript for observation and action elsewhere in this paper indicates the index of the agent).

Overall, our framework first trains on several source tasks $\{\mathcal{S}_i\}$ and then transfers the learned cooperative knowledge to unseen tasks $\{\mathcal{T}_j\}$. As shown in Fig. 1, our learning framework achieves this by designing modules for (1) task representation learning and (2) policy learning. In the following sections, we first introduce how we design the representation learning module and its learning scheme in different phases. Then, we describe the details of policy learning, including the population-invariant structure for dealing with inputs and outputs of varying sizes.

### 2.1 TASK REPRESENTATION LEARNING

Our main idea in achieving knowledge transfer among multi-agent tasks is to capture and exploit both the common structure and the unique characteristics of tasks by learning task representation. A task distinguishes itself from other tasks by its transition and reward functions. Therefore, we incorporate

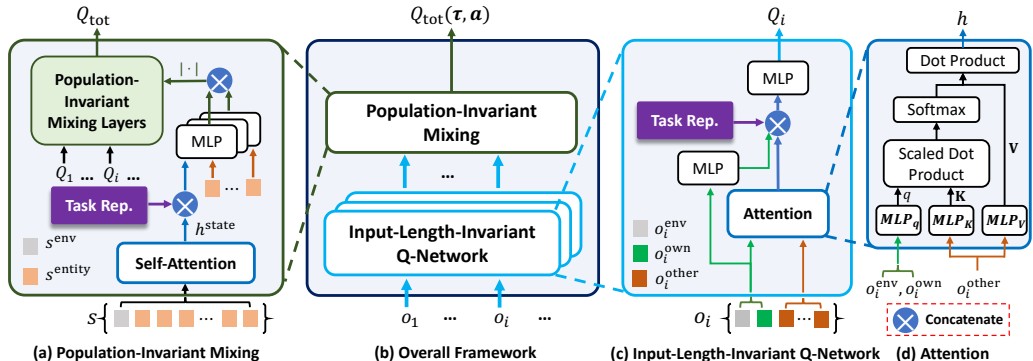

Figure 2: Population-invariant network structure for policy learning.

our task representation learning component into the learning of a forward model that predicts the next state, local observations, and reward given the current state, local observations, and actions.

We associate each task $i$ with a representation $z_i \in \mathbb{R}^m$ and expect it to reflect the relationship of the transition dynamics of tasks. For modeling task similarity, all source and unseen tasks share a representation explainer, which takes as input the task representations and outputs the parameters of the forward model. We then train the representation explainer on all source tasks. Concretely, for a source task $\mathcal{S}_i$, parameters of the forward model are generated as $\eta_i = f_\theta(z_i)$, where $\theta$ denotes the parameters of the representation explainer. The forward model contains three predictors: $f_{\eta_i}^s$, $f_{\eta_i}^o$, and $f_{\eta_i}^r$. Given the current state $s$, agent $j$'s observation $o_j$, and action $a_j$, these predictors estimate the next state $s'$, the next observation $o_j'$, and the global reward $r$, respectively.

A possible method for training on source tasks is to backpropagate the forward model's prediction error to update both the representation explainer and task representations. However, in practice, this scheme leads to representations with very small norms, making it difficult to get a meaningful representation space. To solve this problem, we propose pre-determining the task representation for each source task and learning the representation explainer by backpropagating the prediction error. Such a method can help form an informative task representation space and build a mapping from task representation space to the space of forwarding model parameters. In practice, we initialize source task representations as mutually orthogonal vectors. Specifically, we first randomly generate vectors in $\mathbb{R}^m$ for source tasks, and then use the Schimidt orthogonalization (Björck, 1994) on these vectors to obtain source task representations. With the pre-defined task representations, the representation explainer is optimized to minimize the following loss function:

$$J(\theta) = \sum_{i=1}^{N_{\text{src}}} J_{\mathcal{S}_i}(\theta),$$

where $N_{\text{src}}$ is the number of source tasks, and

$$J_{\mathcal{S}_i}(\theta) = \mathbb{E}_{\mathcal{D}} \left[ \sum_j \left[ \|f_{\eta_i}^s(s, o_j, a_j) - s'\|^2 + \lambda_1 \|f_{\eta_i}^o(s, o_j, a_j) - o_j'\|^2 + \lambda_2 (f_{\eta_i}^r(s, o_j, a_j) - r)^2 \right] \right]$$

is the per-task prediction loss. Here, $\mathcal{D}$ is the replay buffer, and $\lambda_1, \lambda_2$ are scaling factors.

We fix the source task representations and learn the representation explainer during the training phase. In the current implementation, the training data here is collected with the uniform random policy; the data is the same for task representation learning in the transfer phase. We find it simple and effective in the experiment, while more efficient approaches may be adopted here. When it comes to the transfer phase, we aim to find a good task representation that can reflect the similarity of the new task to source tasks. To achieve this goal, we fix the trained representation explainer and learn the task representation by minimizing the prediction loss of the forward model on the new task. Specifically, we randomly initialize a task representation $z$, keep $\theta$ fixed, and get the forward model parameterized by $\eta = f_\theta(z)$. Then the task representation $z$ is updated by backpropagating the prediction loss for transition and reward functions through the fixed $f_\eta$.

To keep the new task representation in the well-formed space of source task representations, we learn the new task representation as a linear combination of source task representations:

$$z = \sum_{i=1}^{N_{\text{src}}} \mu_i z_i \text{ s.t. } \mu_i \geq 0, \sum_{i=1}^{N_{\text{src}}} \mu_i = 1.$$

In this way, what we are learning is the weight vector $\mu$. To make the learning more stable, we additionally optimize an entropy regularization term $\mathcal{H}(\mu)$. The final loss function for learning $z$ is:

$$J_{\mathcal{T}}(\mu) = \lambda \mathcal{H}(\mu) + \mathbb{E}_{\mathcal{D}} \Big[ \sum_j \big[ \|f_\eta^s(s, o_j, a_j) - s'\|^2 + \lambda_1 \|f_\eta^o(s, o_j, a_j) - o_j'\|^2 + \lambda_2 (f_\eta^r(s, o_j, a_j) - r)^2 \big] \Big].$$

The detailed architectures for task representation learning are described in App. F.

## 2.2 TASK POLICY LEARNING

After the task representation is learned by modeling the transition and reward functions, it will be used to learn and transfer the policy on the source and unseen tasks.

A difficulty faced by multi-agent transfer learning is that the dimensions of inputs and outputs vary across tasks. We use a population-invariant network (PIN, Fig. 2) to solve this problem. The idea is not novel, and our contribution here is a light-weight structure that achieves comparable or better performance to other complex PIN modules (Iqbal et al., 2021). The single-task experiments in App. H demonstrate its advantage as the performance gains mainly come from our PIN design. Our PIN uses the value decomposition framework and consists of two main components: an individual Q-network shared by the agents, and a monotonic mixing network (Rashid et al., 2018) that learns the global Q-value as a combination of local Q-values.

For the individual Q-network, like in previous work (Iqbal et al., 2021), we decompose the observation $o_i$ into parts relating to the environment $o_i^{\text{env}}$, agent $i$ itself $o_i^{\text{own}}$, and other entities $o_i^{\text{other}} = \{o_i^{\text{other}_{j \neq i}}\}$. We adopt attention mechanism to get a fixed-dimensional embedding $h$:

$$q = \text{MLP}_q([o_i^{\text{env}}, o_i^{\text{own}}]), \quad \mathbf{K} = \text{MLP}_K([o_i^{\text{other}_1}, \ldots, o_i^{\text{other}_j}, \ldots]),$$

$$\mathbf{V} = \text{MLP}_V([o_i^{\text{other}_1}, \ldots, o_i^{\text{other}_j}, \ldots]), \quad h = \text{softmax}(q\mathbf{K}^{\text{T}}/\sqrt{d_k})\mathbf{V},$$

where $[\cdot, \cdot]$ is the vector concatenation operation, $d_k$ is the dimension of the query vector, and bold symbols are matrices. Embedding $h$ is then fed into the subsequent network together with the task representation $z$ for estimating action values. For the mixing network, we decompose the state $s$ into parts relating to different entities $s^{\text{entity}}$ and the environment $s^{\text{env}}$. We apply a similar self-attention module to integrate information from these parts of the state and obtain a fixed-dimensional embedding vector $h^{\text{state}}$. $h^{\text{state}}$, the task representation $z$, and $s^{\text{entity}}$ are used to generate the parameters of the mixing network. In some multi-agent tasks, the number of actions also varies in different tasks. We use a mechanism similar to other popular population-invariant networks (Wang et al., 2020b; Hu et al., 2021) to deal with this issue, which is discussed in detail in App. G.

During the whole process of policy learning, we fix the task representation $z$. Compared to policy learning, which typically lasts for 2M timesteps, the training of task representation costs few samples. In practice, we collect 50K samples for learning task representations and the representation explainer. When **transferring to new tasks**, we use the individual Q-network and the representation explainer trained on source tasks. We learn the task representation for 50K timesteps and insert it into the individual Q-network. Agents execute in a decentralized manner according to this Q-network.

## 3 CASE STUDY

To offer an intuitive understanding of the task representation learning, we design a simple case study to show how MATTAR works, which refers to a two-player navigation problem. In this experiment setting, each task corresponds to a goal position and two agents are expected to cooperatively navigate to the specific goal from an initial region. We design eight source tasks with different goals and one unseen target task. Detailed descriptions of the experiment setting can be found in App. B.

In the training phase, we learn the representation explainer and the agent policy on eight source tasks. Afterward, we optimize the task representation on the unseen task and transfer the learned policy. In Fig. 3(a), we study the learned representation for the unseen target task. We can find source tasks 4 and 3 occupy the largest two coefficients, which are about 70% and 30%, respectively. Actually, these two source tasks are more relevant, as their goal positions are closer to that of the target unseen task compared to other source tasks, and our mechanism for task representation learning captures the similarity between their reward functions. To confirm our hypothesis, we embed the learned task

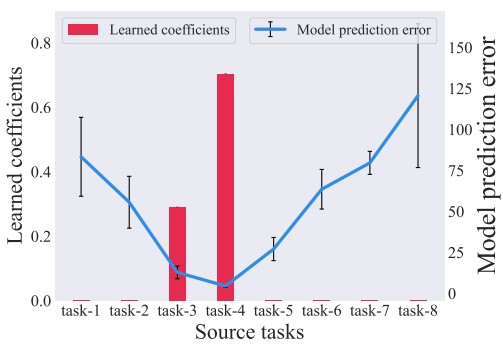

(a) The learned task representations.

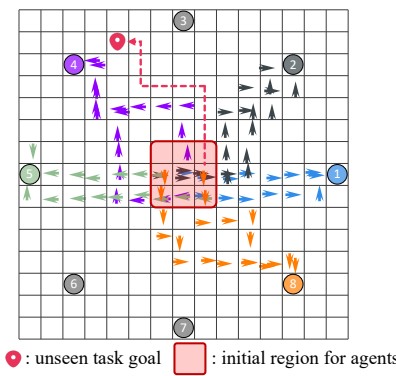

(b) Visualization of the task trajectories.

Figure 3: The results of the case study on the two-player navigation problem. (a) The task representation learned by MATTAR. (b) We draw roll-outed trajectories of agent policy on source tasks (represented by circular symbols) by arrows, with different colors for different tasks. For simplicity, we only selectively show trajectories for five source tasks. Besides, we test MATTAR on one unseen task and draw the trajectory with continuous red dashed arrow. More details can be found in App. B.

representation into the representation explainer and test the prediction error of the obtained forward model in each source task. The results exhibit that lower prediction errors are obtained in tasks 4 and 3, implying that the learned representation can reflect different tasks (e.g., the rewards in these tasks).

Furthermore, we visualize the task trajectories roll-outed in Fig. 3(b). Capturing the similarity between the unseen target task and source tasks by task representation, the transferred policy reuses the policy in relevant source tasks. When embedding different source task representations, the agent policy exhibits different behavioral strategies to reach the corresponding goal. When encountered an unseen new task, the learned task representation guides the agent to move up first, which is typical behavior in task 4, and further to the target position when the goal is within sight, demonstrating MATTAR can indeed extract common latent representation among tasks to ease solving unseen tasks.

## 4 EXPERIMENTS

In this section, we design experiments to evaluate the following properties of the proposed method: (1) Generalizability to unseen tasks. Our learning framework can extract knowledge from multiple source tasks and transfer the cooperation knowledge to unseen tasks, and task representations play an indispensable role in transfer (Sec. 4.1). (2) A good initialization for policy fine-tuning. Fine-tuning the transferred policy can succeed in super hard tasks, which can not be solved when learning from scratch (Sec. 4.2). (3) Benefits of multi-task training. Our multi-task learning paradigm helps the model better leverage knowledge of different source tasks to boost the learning performance compared to training on source tasks individually (Sec. 4.3). (4) Performance advantages on single tasks. Although we did not design our framework for single-task learning, our network performs better against the underlying algorithm (App. H).

We evaluate MATTAR on the SMAC (Samvelyan et al., 2019) and MPE (Lowe et al., 2017) benchmarks. To better fit the multi-task training setting, we extend the original SMAC maps and sort out three task series, which involve various numbers of Marines, Stalkers/Zealots, and Marines/Maneuvers/Medivacs, respectively. For the MPE tasks, we test on a discrete version of `Spread` and `Gather`, with different numbers of agents trying to cover all or one landmarks. The detailed environmental settings are described in App. C. To ensure fair evaluation, we carry out all the experiments with five random seeds, and the results are shown with a $95\%$ confidence interval in all figures. For a more comprehensive description of experimental details, please refer to App. E.

### 4.1 GENERALIZABILITY TO UNSEEN TASKS

As the major desiderata of our method, we expect that MATTAR has better transfer performance on unseen tasks. We note that there are few related works applicable to tasks of varying sizes, and we

Table 1: Transfer performance (mean win rates with variance) on the first series of SMAC maps.

| | Source Tasks | | | Unseen Tasks | | | | |
|---|---|---|---|---|---|---|---|---|
| | 2s3z | 3s5z | 3s5z_3s6z | 1s8z | 1s9z | 2s8z | 2s9z | 7s3z |
| MATTAR | **1.00**±0.00 | **0.99**±0.01 | **0.48**±0.13 | **0.79**±0.09 | **0.60**±0.12 | **0.93**±0.09 | **0.84** ±0.04 | **0.16**±0.12 |
| w/o task rep. | 0.99±0.01 | 0.96±0.02 | 0.20±0.08 | 0.12±0.13 | 0.07±0.12 | 0.47±0.18 | 0.25±0.20 | 0.15±0.19 |
| 0 task rep. | 0.23±0.12 | 0.08±0.05 | 0.00±0.00 | 0.02±0.03 | 0.00±0.00 | 0.01±0.01 | 0.02±0.02 | 0.00±0.00 |
| UPDeT-b | 0.94±0.04 | 0.86±0.13 | 0.09±0.08 | 0.16±0.11 | 0.11±0.10 | 0.29±0.22 | 0.15±0.13 | 0.02±0.04 |
| UPDeT-m | 0.60±0.11 | 0.47±0.15 | 0.03±0.03 | 0.08±0.06 | 0.04±0.04 | 0.14±0.12 | 0.06±0.05 | 0.01±0.01 |
| REFIL | 0.75±0.09 | 0.43±0.13 | 0.01±0.01 | 0.08±0.04 | 0.03±0.01 | 0.08±0.05 | 0.05±0.04 | 0.06±0.04 |

Table 2: Transfer performance (mean win rates with variance) on the second series of SMAC maps.

| | Source Tasks | | | Unseen Tasks | | | | |
|---|---|---|---|---|---|---|---|---|
| | MMM | MMM2 | MMM4 | MMM0 | MMM1 | MMM3 | MMM5 | MMM6 |
| MATTAR | **1.00**±0.00 | **0.92**±0.20 | **0.93**±0.12 | **0.98**±0.02 | **0.97**±0.04 | **0.86**±0.10 | **0.47**±0.15 | **0.09**±0.02 |
| w/o task rep. | 0.94±0.05 | 0.23±0.39 | 0.33±0.25 | 0.81±0.15 | 0.37±0.36 | 0.07±0.05 | 0.22±0.30 | **0.09**±0.17 |
| 0 task rep. | 0.61±0.07 | 0.07±0.06 | 0.21±0.22 | 0.28±0.19 | 0.11±0.13 | 0.08±0.10 | 0.08±0.12 | 0.02±0.04 |
| UPDeT-b | **1.00**±0.00 | 0.78±0.04 | 0.41±0.14 | 0.73±0.21 | 0.84±0.07 | 0.57±0.15 | 0.00±0.00 | 0.00±0.00 |
| UPDeT-m | 0.48±0.03 | 0.15±0.19 | 0.20±0.07 | 0.30±0.16 | 0.27±0.13 | 0.28±0.08 | 0.00±0.00 | 0.00±0.00 |
| REFIL | 0.97±0.01 | 0.04±0.02 | 0.06±0.03 | 0.93±0.02 | 0.38±0.06 | 0.12±0.04 | 0.00±0.00 | 0.00±0.00 |

Table 3: Transfer performance (mean win rates with variance) on the third series of SMAC maps.

| | Source Tasks | | | | Unseen Tasks | | | |
|---|---|---|---|---|---|---|---|---|
| | 5m | 5m_6m | 8m_9m | 10m_11m | 3m | 4m | 4m_5m | 6m |
| MATTAR | **1.00**±0.00 | 0.72±0.05 | **0.83**±0.05 | 0.81±0.09 | **0.94**±0.27 | **0.97**±0.02 | 0.04±0.05 | **1.00**±0.00 |
| w/o task rep. | 0.97±0.01 | 0.01±0.02 | 0.01±0.01 | 0.01±0.03 | 0.86±0.03 | 0.88±0.04 | 0.00±0.00 | 0.95±0.03 |
| 0 task rep. | 0.78±0.39 | 0.16±0.12 | 0.30±0.24 | 0.40±0.28 | 0.00±0.00 | 0.21±0.15 | 0.01±0.01 | 0.67±0.47 |
| UPDeT-b | **1.00**±0.00 | **0.93**±0.05 | 0.81±0.19 | **0.94**±0.04 | 0.81±0.08 | 0.95±0.06 | **0.29**±0.17 | **1.00**±0.00 |
| UPDeT-m | 0.77±0.09 | 0.32±0.03 | 0.35±0.05 | 0.43±0.02 | 0.36±0.04 | 0.57±0.03 | 0.10±0.06 | 0.91±0.09 |
| REFIL | 0.73±0.03 | 0.00±0.00 | 0.01±0.01 | 0.03±0.02 | 0.68±0.06 | 0.74±0.02 | 0.00±0.00 | 0.71±0.02 |
| | Unseen Tasks | | | | | | | |
| | 6m_7m | 7m | 7m_8m | 8m | 9m | 9m_10m | 10m | 10m_12m |
| MATTAR | 0.74±0.15 | **1.00**±0.00 | **0.83**±0.04 | **1.00**±0.00 | **1.00**±0.00 | **0.84**±0.09 | **1.00**±0.00 | **0.07**±0.01 |
| w/o task rep. | 0.03±0.02 | 0.94±0.03 | 0.08±0.10 | 0.93±0.04 | 0.86±0.05 | 0.04±0.02 | 0.52±0.22 | 0.00±0.00 |
| 0 task rep. | 0.31±0.22 | 0.67±0.47 | 0.49±0.35 | 0.67±0.47 | 0.66±0.46 | 0.32±0.24 | 0.65±0.46 | 0.00±0.00 |
| UPDeT-b | **0.78**±0.05 | 0.99±0.01 | 0.73±0.11 | 0.99±0.02 | 0.99±0.01 | 0.80±0.16 | 0.99±0.01 | **0.07**±0.04 |
| UPDeT-m | 0.35±0.10 | 0.92±0.03 | 0.38±0.05 | 0.83±0.05 | 0.66±0.11 | 0.33±0.09 | 0.17±0.08 | 0.03±0.02 |
| REFIL | 0.01±0.00 | 0.66±0.03 | 0.01±0.01 | 0.63±0.05 | 0.55±0.05 | 0.01±0.00 | 0.46±0.02 | 0.00±0.00 |

compare our method against the state-of-the-art multi-agent transfer method UPDeT (Hu et al., 2021) and REFIL (Iqbal et al., 2021). UPDeT transfers knowledge from a single source task. For a fair comparison, we transfer from each source task to every unseen task and calculate the best (UPDeT-b) and mean (UPDeT-m) performance on each unseen task. For the test phase, we conduct transfer experiments on both source tasks and unseen tasks. Results on the SMAC benchmark are shown in Tab. 1∼3, and those on the MPE are shown in Tab. 4 and 5.

We find that MATTAR shows superior transfer performance on unseen tasks, significantly outper-forming UPDeT-b and REFIL, especially in complex settings requiring sophisticated coordination like the MMM series. MATTAR is effective even for unseen tasks that are very different from source tasks. For example, the MATTAR policy learned on 2s3z, 3s5z, and 3s5z_vs_3s6z can win 79% of games on 1s8z. On MPE tasks, MATTAR also shows good generalization performance on unseen tasks with different numbers of agents and overall outperforms the baselines and ablations. For example, On Spread tasks, MATTAR exhibits significant advantage over UPDeT on all unseen tasks and outperforms REFIL with a margin of 13% on the unseen task with 6 agents.

To explain the performance of our method, we first check the learned representations on unseen tasks and then carry out ablation studies.

**What representations are learned for unseen tasks?** When encountering an unseen task, we first learn its representation as a linear combination of the representations for source tasks. Specifically, we directly update the coefficients of this linear combination by backpropagating the prediction error of the forward model. For a deeper understanding of how our method transfers the learned knowledge, here we investigate the learned coefficients of the linear combination.

Table 4: Evaluation on MPE: Transfer performance (mean success rates with variance) on `Spread` with different numbers of agents.

| | Source Tasks | | | | Unseen Tasks | | | |
|---|---|---|---|---|---|---|---|---|
| | 2 | 4 | 7 | 9 | 3 | 5 | 6 | 8 |
| MATTAR | **1.00**±**0.00** | **0.97**±0.03 | **0.17**±0.16 | **0.12**±0.05 | **0.98**±0.01 | **0.75**±0.12 | **0.19**±0.10 | **0.09**±0.09 |
| w/o task rep. | **1.00**±0.00 | 0.55±0.30 | 0.02±0.01 | 0.00±0.00 | 0.83±0.13 | 0.30±0.13 | 0.03±0.04 | 0.00±0.00 |
| 0 task rep. | 0.98±0.01 | 0.96±0.03 | 0.13±0.18 | 0.09±0.04 | 0.94±0.07 | 0.70±0.17 | 0.09±0.09 | **0.09**±0.07 |
| UPDeT-b | 0.73±0.23 | 0.09±0.00 | 0.02±0.02 | 0.00±0.00 | 0.41±0.16 | 0.05±0.02 | 0.00±0.00 | 0.02±0.02 |
| UPDeT-m | 0.39±0.12 | 0.06±0.00 | 0.00±0.00 | 0.00±0.00 | 0.18±0.08 | 0.02±0.01 | 0.00±0.00 | 0.01±0.01 |
| REFIL | **1.00**±**0.00** | 0.93±0.04 | 0.07±0.04 | 0.01±0.01 | 0.96±0.03 | 0.73±0.06 | 0.06±0.05 | 0.05±0.00 |

Table 5: Evaluation on MPE: Transfer performance (mean success rates with variance) on `Gather` with different numbers of agents.

| | Source Tasks | | | | Unseen Tasks | | | |
|---|---|---|---|---|---|---|---|---|
| | 2 | 4 | 7 | 9 | 3 | 5 | 10 | 15 |
| MATTAR | **1.00**±0.00 | **1.00**±0.00 | 0.82±0.11 | 0.84±0.17 | **1.00**±0.00 | **1.00**±0.00 | 0.63±0.12 | **0.55**±0.22 |
| w/o task rep. | **1.00**±0.00 | **1.00**±0.00 | 0.88±0.09 | 0.84±0.17 | **1.00**±0.00 | 0.99±0.01 | 0.62±0.13 | 0.53±0.18 |
| 0 task rep. | **1.00**±0.00 | **1.00**±0.00 | 0.73±0.26 | 0.75±0.18 | **1.00**±0.00 | 0.99±0.01 | 0.54±0.26 | 0.50±0.29 |
| UPDeT-b | 0.66±0.34 | 0.38±0.38 | 0.25±0.12 | 0.09±0.09 | 0.61±0.36 | 0.36±0.27 | 0.03±0.03 | 0.06±0.06 |
| UPDeT-m | 0.32±0.16 | 0.15±0.15 | 0.07±0.02 | 0.04±0.04 | 0.20±0.10 | 0.11±0.07 | 0.01±0.01 | 0.02±0.02 |
| REFIL | **1.00**±0.00 | **1.00**±0.00 | **0.99**±0.02 | **0.99**±0.01 | **1.00**±1.00 | **1.00**±0.00 | **0.73**±0.14 | 0.51±0.14 |

In Tab. 6, we show the coefficients of two unseen tasks for each series of maps. We observe that the largest coefficient typically corresponds to the source task that is the most similar to the unseen task. For example, 5m is the closest source task to the unseen task 4m, and the coefficient of 5m is $61\%$. There are also some exceptions. For example, for the unseen task 10m_vs_12m, the coefficients of two source tasks, 5m and 10m_vs_11m, are equal, and they together take up $86\%$ of all the coefficients. While 10m_vs_11m is very similar to 10m_vs_12m as the team composition is similar, the policy for solving 5m is different from that for 10m_vs_12m, we find that agents first form some groups to set up an attack curve quickly in 10m_vs_12m. Therefore, on a local battlefield, there are around five allies fighting against a similar number of enemies. In this case, the policy learned from 5m can be used locally to promote coordination in 10m_vs_12m.

We conclude that, in MATTAR, task representation learning captures the similarity of task dynamics and thereby discovers opportunities of policy reuse. These representations help unseen tasks effectively leverage the knowledge from the most similar source tasks or reuse the knowledge from a mixing of source tasks, which is also consistent with the results in case study (Sec.3).

**Ablations**. To further investigate the role of task representations in our method, we design two ablations. (1) 0 `task rep.` uses the trained MATTAR models but feed a zero-valued task representation into the policy network when transferring. (2) `w/o task rep.` denotes completely remove components related to task representation, including the forward model, the representation learning process, and the task representation in the policy network. Train and transfer the population-invariant policy. This ablation can reveal the generalization ability of the policy network itself.

Results are shown in Tab. 1∼3. We can see that these ablations bring about a large drop in the performance of MATTAR on nearly all unseen tasks. For example, after trained on 2s3z, 3s5z, and 3s5z_vs_3s6z, MATTAR achieves a win rate of $0.79$ on 1s8z, while `w/o task rep.` obtains a win rate of $0.12$ and 0 `task rep.` does not win any games. We can thus conclude that **task representations play an indispensable role in policy transfer**.

### 4.2 A GOOD INITIALIZATION FOR POLICY FINE-TUNING

When evaluating the performance of MATTAR on unseen tasks, we only train the task representations and reuse other parts of the policy. In this section, we investigate the performance of MATTAR after fine-tuning the policy. Specifically, on an unseen task, we first fix the representation explainer and train the task representation $z$ for 50K timesteps. Then we fix the learned task representation and fine-tune the policy for 2M timesteps. We compare against learning from scratch in Fig. 4. Learning from scratch means the policy is randomly initialized and trained for 2M timesteps. We consider two versions for learning from scratch. The first version (`Learn from scratch w/o repr.`) totally removes

Table 6: **Our method models task relationship and exploits it for knowledge transfer.** Task representations of unseen tasks are learned as a linear combination of source tasks' representations, whose coefficients are shown in this table. These weights reveal the encoded task relationship.

| Source | Unseen Tasks | |
|---|---|---|
| | 4m | 10m_12m |
| 5m | **0.61** | **0.43** |
| 5m_6m | 0.13 | 0.07 |
| 8m_9m | 0.14 | 0.08 |
| 10m_11m | 0.12 | **0.43** |

| Source | Unseen Tasks | |
|---|---|---|
| | 3s4z | 3s5z_3s7z |
| 2s3z | 0.21 | 0.18 |
| 3s5z | **0.59** | 0.21 |
| 3s5z_3s6z | 0.21 | **0.61** |

| Source | Unseen Tasks | |
|---|---|---|
| | MMM0 | MMM6 |
| MMM | **0.44** | 0.30 |
| MMM2 | 0.15 | 0.14 |
| MMM4 | 0.40 | **0.56** |

(a) 10m_vs_12m

(b) MMM6

(c) 3s5z_vs_3s7z

Figure 4: On unseen tasks: task representations provide a good initialization. Fine-tuning the policy can learn cooperation policies effectively than learning from scratch.

task representations from the policy, while the second version (`Learn from scratch w/ repr.`) inserts the same task representation $z$ as in the fine-tuning experiments.

We observe that the task representation and the reused policy provide a good initialization. For example, on `10m_vs_12m`, after fine-tuning for 2M timesteps, `MATTAR fine-tuning` converges to a win rate of around $0.86$, while both versions of learning from scratch can only achieve a win rate of about $0.4$. Furthermore, empirically only `MATTAR fine-tuning` achieves non-zero winning rates on `3s5z_vs_3s7z`, a task harder than the super hard map `3s5z_vs_3s6z`.

## 4.3 BENEFITS OF MULTI-TASK LEARNING

MATTAR adopts a scheme where multiple sources are learned simultaneously. Our aim is to leverage knowledge from more tasks and to be able to generalize the learned knowledge to a larger set of unseen tasks. Empirically, we find that this multi-task training setting helps not only unseen tasks but also the source tasks themselves.

In Fig. 5, we present the performance of MATTAR on source tasks when training with multiple and a single source task. The experiments are carried out on three tasks from three different series. We can see that training on multiple tasks significantly boosts learning performance. For example, on `3s5z_vs_3s6z`, after 2M training samples, MATTAR with multiple tasks converges to the win rate of around $0.6$, while training solely on this task can only achieve a win rate of about $0.05$. Furthermore, we can also observe that MATTAR significantly outperforms the state-of-the-art deep multi-agent multi-task method (REFIL (Iqbal et al., 2021)). These results reveal that MATTAR also provides a good learning framework for multi-agent multi-task learning. It can leverage experience on other tasks to improve performance on a similar task.

## 5 RELATED WORK

**Multi-task learning with metadata** (You et al., 2016; Zheng et al., 2019) approaches have been explored to exploit richer context information in the form of learned task embeddings, with a focus on task relation discovery. Sodhani et al. (2021) use context information in high-level, under-specified, and unstructured natural language for multi-task learning. Compared to the settings studied in this paper, multi-agent problems pose additional challenges. Transition dynamics and reward functions in multi-agent tasks are influenced by the interaction among all agents so the relation between tasks has

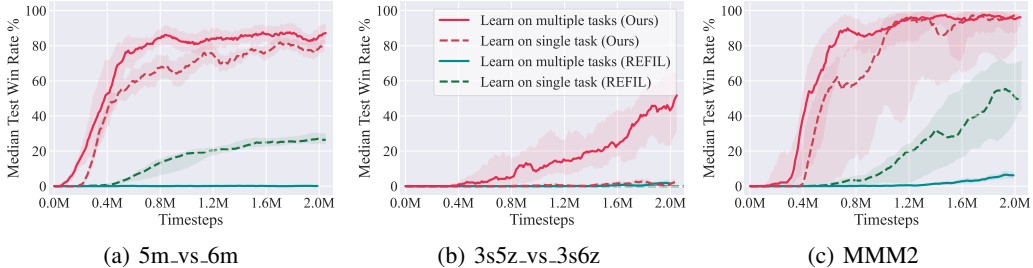

Figure 5: On source tasks: MATTAR provides a framework for multi-agent multi-task learning. Training on multiple tasks helps improve performance than learning on a single task.

to be conditioned on the interaction pattern of multiple agents. Different from previous contextual multi-task RL methods, we include agent interaction modeling in the task representation learning module to deal with this problem. Nevertheless, we think incorporating natural language as prior high-level metadata of tasks can further improve the performance of our method in further work.

Related to our work, **context-based meta-RL** also reuses experience from multiple tasks and infers how to solve a new task from collected small amounts of experience (Zintgraf et al., 2021). For example, PEARL (Rakelly et al., 2019) performs online task inference by probabilistic filtering of latent task variables. Other meta-RL methods exploit the dynamics of recurrent networks for implicit task inference and fast adaptation (Wang et al., 2017; Duan et al., 2016). These methods tend to form dynamic variables or embeddings, while the context information is utilized to obtain fixed task representations in our approach. Besides, although their ability of task adaptation has been proved, context-based meta-RL has not been extended to multi-agent cases and cannot deal with varying numbers of actions and different lengths of observation inputs. For future work, it is important to discuss whether the inspiration of context-based meta-RL can further improve the performance of our multi-agent multi-task learning framework.

**Learning task or skill embeddings** for multi-task transfer reinforcement learning has also been extensively explored. When testing, Hausman et al. (2018) learn a new skill embedding by inter-polating learned skill embeddings on source tasks by approximate variational inference. Arnekvist et al. (2019) also focus on learning skill embeddings but is conditioned on optimal Q-functions for different skills. Co-Reyes et al. (2018) adopt a hierarchical framework and learns a high-level policy to control a low-level skill latent space. Similar to ours, Co-Reyes et al. (2018) learn the low-level skill space by encoding experience trajectories and decoding states and actions. Another work that is similar to ours is Zhang et al. (2018), which conditions the policy on latent representations learned by dynamics and reward module. Lan et al. (2019) learn task embeddings, fixes the policy, and updates the encoder at the test time. Compared to these methods, our task representation learning method differs by (1) it is specially designed for multi-agent settings and considers varying input sizes and intra-agent interaction; and (2) to the best of our knowledge, the alternatively fixed learning scheme is proposed for the first time. More contents include **multi-task reinforcement learning, multi-agent transfer learning, multi-agent representation learning, and modular RL** are shown in App. A.

## 6 CONCLUSION

In this paper, we study cooperative multi-agent transfer reinforcement learning by learning task representations that model and exploit task relationship. Previous work on multi-agent transfer learning mainly deals with the varying population and input lengths, relying on the generalization ability of neural networks for cooperation knowledge transfer, ignoring the task relationship among tasks. Our method improves the transfer performance by learning task representations that capture the difference and similarities among tasks. When facing a new task, our approach only needs to obtain a new representation before transferring the learned knowledge. Taking advantage of task relationship mining, our proposed method MATTAR achieves the best transfer performance and other bonuses compared with multiple baselines. An important direction in the future is the transfer among tasks from different task distributions, and whether a linear combination of source tasks' representations can fully represent unseen tasks is also a valuable topic.

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

Appendix

## A  More Related Work

**Transfer learning**  Transfer learning holds the promise of improving the sample efficiency of reinforcement learning (Zhu et al., 2020), which learns knowledge from source tasks to accelerate the learning efficiency in the unseen task. Previous work uses successor features, decouples the transition dynamics and reward function, and learns faster in simulated navigation and robotic arm settings (Barreto et al., 2018). DSE (Petangoda et al., 2019) models the transfer process as variational inference and further learns a latent space to transfer skills across different dynamics. Sun et al. (2022) apply a model-based regularizer to learn task-level transfer across various observation spaces. Related to our work, REPAINT (Tao et al., 2021) combines task representations with on-policy learning and uses an advantage-based experience selection approach to transfer useful samples. Our method differs from these works by (1) its mechanism for alternately updating the task representation and the representation explainer and (2) its mechanism for handling different input/output sizes.

The basic idea behind **multi-agent transfer learning** (Silva & Costa, 2021) is to reuse knowledge from other tasks or other learning agents, corresponding to intra-agent transfer and inter-agent transfer, respectively. It is expected that the knowledge reuse can accelerate coordination compared to learning from scratch. The inter-agent transfer paradigm aims at reusing knowledge from other agents with different sensors or (possibly) internal representations via communication. DVM (Wadhwania et al., 2019) treats the multi-agent problem as a multi-task problem to combine knowledge from multiple tasks and then distills the knowledge by a value matching mechanism. LeCTR (Omidshafiei et al., 2019) learns to teach in a multi-agent environment and learns to advise others in a peer-to-peer manner. MAPTF (Yang et al., 2021) takes a further step by proposing an option-based policy transfer for multi-agent cooperation, and it significantly boosts the performance of existing methods in both discrete and continuous state spaces. On the other hand, intra-agent transfer refers to reusing knowledge from previous tasks, focusing on transferring knowledge across multi-agent tasks. The varying populations and input lengths impede the transfer among agents, with which the graph neural networks and the transformer play promising roles. DyMA-CL (Wang et al., 2020c) designs various transfer mechanisms across curricula to accelerate the learning process based on a dynamic agent-number network. EPC (Long et al., 2019) proposes a curriculum learning paradigm via an evolutionary approach to scale up the population number of agents. UPDeT (Hu et al., 2021) and PIT (Zhou et al., 2021) make use of the generalization ability of the transformer to accomplish the multi-agent cooperation and transfer between tasks. Although these methods can accelerate the learning efficiency of MARL algorithms, they do not exploit task similarity for better transfer performance. By contrast, our method explicitly models task relationship by learning a hidden space in which tasks with similar dynamics have similar representations.

**Multi-task reinforcement learning** is another relevant research topic that enables an RL agent to leverage experience from multiple tasks to improve sample efficiency and avoid learning from scratch on every single task. Various approaches have been proposed to achieve multi-task learning, such as distilling separate tasks' knowledge into a shared policy (Levine et al., 2016; Ghosh et al., 2018; Xu et al., 2020), conditioning policies on tasks (Deisenroth et al., 2014), mapping tasks to parameters of a policy (da Silva et al., 2012; Kober et al., 2012; Yang et al., 2020), and solving the problem of negative interference (Du et al., 2018; Suteu & Guo, 2019; Yu et al., 2020) meaning that gradients of different tasks may conflict.

**Multi-agent representation learning**  Learning effective representation in MARL is receiving significant attention for its effectiveness in solving many important problems. CQ-Learning (Hauwere et al., 2010) learns to adapt the state representation for multi-agent systems to coordinate with other agents. Grover et al. (2018) learn useful policy representations to model agent's behavior in a multi-agent system. LILI (Xie et al., 2020) learns latent representations to capture the relationship between ego-agent's behavior and the other agent's future strategy. RODE (Wang et al., 2021b) uses an action encoder to learn action representations and applies clustering methods to decompose the joint action space into restricted role action spaces to reduce the policy search space of multi-agent cooperation. MAR (Zhang et al., 2021b) learns meta representation for multi-agent generalization. Our approach differs from these works by learning representations for tasks and using a representation explainer for efficient policy transfer.

Another line of research on single-agent multi-task learning that has the potential to be effective in multi-agent settings is **modular RL**. Modular RL decentralizes the control of multi-joint robots by learning policies for each actuator and thus holds the promise to deal with input and output with varying lengths. Each joint has its controlling policy, and they coordinate with each other via various message passing schemes. To do so, Wang et al. (2018) and Pathak et al. (2019) represent the robot's morphology as a graph and use GNNs as policy and message passing networks. Huang et al. (2020) use both bottom-up and top-down message passing schemes through the links between joints for coordinating. All of these GNN-like works show the benefits of modular policies over a monolithic policy in tasks tackling different morphologies. However, recently, Kurin et al. (2021) validated a hypothesis that any benefit GNNs can extract from morphological structures is outweighed by the difficulty of message passing across multiple hops. They further propose a transformer-based method, AMORPHEUS, that utilizes self-attention mechanisms as a message passing approach. Although modular RL can deal with varying action numbers of different tasks and can implicitly model the interaction between agents through GNN, Transformer, or message passing, it still cannot cope with varying observation lengths and does not incorporate task-level context information compared to our method. In summary, our work distinguishes itself from previous multi-task work by (1) its flexibility of handling varying lengths of observation and varying numbers of actions; (2) its utilization of agent-interaction modeling in capturing task relation; and (3) its alternatively fixed task representation learning scheme.

## B    DETAILS ABOUT CASE STUDY

**Problem definition.** In case study (Sec. 3), we design a two-player navigation problem, which essentially is a grid-world problem. The whole map is a grid world with size $[15, 15]$, and in each episode two agents are randomly initialized in the centering initial region which is a $3 \times 3$ sub-area. The agents can observe local information within a sight range of $1$ and are expected to reach a specific goal position for the current task. Each task corresponds to a specific goal position and there exist eight source tasks as marked in Fig. 6.

For the reward function, we define the reward $r_t$ at each timestep $t$ as the opposite of the $L_1$ distance from the agents to the target position:

$$r_t = -\sum_{i=1}^{2} \left[ \texttt{abs}(agent\_pos_i[0] - goal\_pos[0]) + \texttt{abs}(agent\_pos_i[1] - goal\_pos[1]) \right].$$

**Application of MATTAR**. Note that the transition functions are all the same in different tasks. To better capture the similarity and distinctions between tasks, we only optimize the reward prediction loss for forward model in the task representation learning step. The training phase is conducted on the eight source tasks, where the representation explainer and the agent policy are trained. When transferring to the unseen task, we learn the task representation by optimizing the reward prediction loss on the new task and insert the learned task representation into the trained policy network to obtain the transferred policy.

**Two results in the main text**. For Fig. 3(a), we conduct transfer phase for multiple times independently. Each time, we record the learned coefficients for the target task, and show the results in the form of a bar chart where each bar corresponds to a specific source task. Besides, each time we obtain a target task representation, we insert it into the representation explainer and obtain a forward model. We test the prediction loss of this forward model on eight

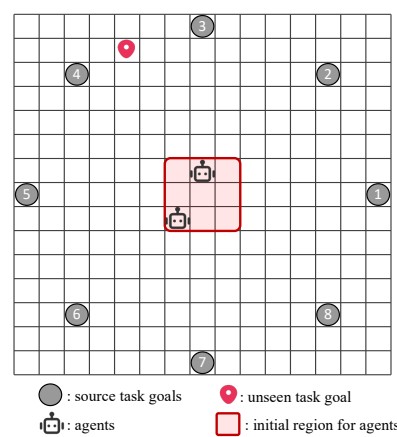

Figure 6: Tow-player Navigation problem for Case Study.

source tasks and present the results in the form of a plot chart. Besides, note that the samples for computing model prediction error are collected with random policies. For Fig. 3(b), we first insert source task representations to the trained agent policy and roll out several trajectories. For simplicity of presentation, we only visualize the trajectories of agent 1 and selectively show trajectories for

five source tasks (1, 2, 4, 5 and 8). We visualize trajectories for two independent runs. To avoid overlapping of arrows, we place an arrow at a random position within a grid to represent that the agent once passed this grid and the direction of arrow indicate the moving direction of agent. For the trajectory of the unseen target task, we visualize one trajectory and represent it with a dashed line.

## C  DETAILS ABOUT THE BENCHMARKS

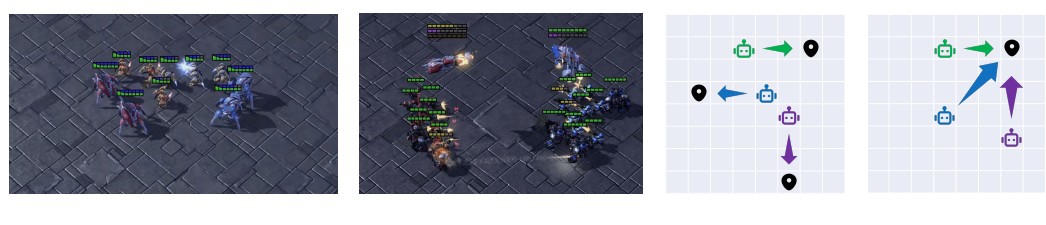

(a) SMAC: 2s3z    (b) SMAC: MMM2    (c) MPE: Spread  (d) MPE: Gather

Figure 7: Snapshots of the experimental environments used in this paper.

**SMAC**   (Fig. 7(a)∼(b)) StarCraft II Micromanagement Benchmark (Samvelyan et al., 2019) contains combat scenarios of StarCraft II unit micromanagement tasks and is a popular benchmark for multi-agent reinforcement learning. We consider a partially observable setting, where an agent can only see a circular area around it with a radius equal to its sight range, which is default to 9. We train ally units with MATTAR to fight against enemy units controlled by the built-in AI. At the beginning of each episode, allies and enemies spawn in pre-defined regions on the map. Every agent takes actions from a discrete action space including `no-op`, `move[direction]`, `attack[enemyid]`, and `stop`. Under the control of these actions, agents can move and attack on a continuous map. Agents will get a shared reward equal to the total damage dealt to enemy units at each timestep. Killing each enemy unit and winning the combat (killing all the enemies) will bring additional bonuses of 10 and 200, respectively. We consider three series of SMAC tasks, each including various maps. The detailed descriptions are shown in Tab. 7∼9.

**MPE**   (Fig. 7(c)∼(d)) Multi-Agent Particle Environment (Lowe et al., 2017) is a multi-agent particle world containing several navigation and communication tasks. In our experiments, we consider a discrete version of MPE and use two tasks, `Spread` and `Gather`, to evaluate our method. In both tasks, there are `n_agent` agents on a field with size `[field_size, field_size]` tasked to reach landmarks. The agents can observe objects around it within a distance of `sight_range`. When a landmark is within a `reach_range× reach_range` sub-field around an agent, we say the agent has reached the landmark. In `Spread`, we require each agent to reach a landmark that is not occupied by any other agents, while in `Gather`, the agents share a common landmark. In both of these tasks, only when all agents reach the landmark, a collective reward of 1 is given. For both `Spread` and `Gather`, we test several tasks with different numbers of agents. The detailed settings of these tasks are listed in Tab. 10 and 11.

## D  NETWORK ARCHITECTURE AND HYPERPARAMETERS

Our implementation of MATTAR is based on PyMARL [1] with StarCraft 2.4.6.2.69232 and uses its default hyperparameter settings. We apply the default $\epsilon$-greedy action selection algorithm to every algorithm, as $\epsilon$ decays from 1 to 0.05 in 50K timesteps. We also adopt typical Q-learning training tricks like the target network and double Q-learning. MATTAR has hyperparameters $\lambda_1, \lambda_2$, and $\lambda$ as the scaling factors of the observation prediction loss, the reward prediction loss, and the entropy regularization term, respectively. We set them to 1, 10, and 0.1 across all experiments. For other hyper-parameters, we use the default settings of QMIX presented in the PyMARL framework. For RODE (Wang et al., 2021b), ASN (Wang et al., 2020b), QPLEX (Wang et al., 2021a), QMIX (Rashid et al., 2018), and UPDeT (Hu et al., 2021), we use the codes provided by the authors with the default

---

[1] We use PyMARL with SC2.4.6.2.6923. Performance is not always comparable among versions.

Table 7: Settings of tasks in the MMM series. The bolded names indicate the source tasks.

| Map Name | Ally Units | Enemy Units | Type | Difficulty |
|---|---|---|---|---|
| MMM0 | 1 Medivac, 2 Marauders, 5 Marines | 1 Medivac, 2 Marauders, 5 Marines | Asymmetric & Heterogeneous | Easy |
| **MMM** | 1 Medivac, 2 Marauders, 7 Marines | 1 Medivac, 2 Marauders, 7 Marines | Asymmetric & Heterogeneous | Easy |
| MMM1 | 1 Medivac, 1 Marauders, 7 Marines | 1 Medivac, 2 Marauders, 7 Marines | Asymmetric & Heterogeneous | Hard |
| **MMM2** | 1 Medivac, 2 Marauders, 7 Marines | 1 Medivac, 3 Marauders, 8 Marines | Asymmetric & Heterogeneous | Super Hard |
| MMM3 | 1 Medivac, 2 Marauders, 8 Marines | 1 Medivac, 3 Marauders, 9 Marines | Asymmetric & Heterogeneous | Super Hard |
| **MMM4** | 1 Medivac, 3 Marauders, 8 Marines | 1 Medivac, 4 Marauders, 9 Marines | Asymmetric & Heterogeneous | Super Hard |
| MMM5 | 1 Medivac, 3 Marauders, 8 Marines | 1 Medivac, 4 Marauders, 10 Marines | Asymmetric & Heterogeneous | Super Hard |
| MMM6 | 1 Medivac, 3 Marauders, 8 Marines | 1 Medivac, 4 Marauders, 11 Marines | Asymmetric & Heterogeneous | Super Hard |

Table 8: Settings of tasks in the SZ series. The bolded names indicate the source tasks.

| Map Name | Ally Units | Enemy Units | Type | Difficulty |
|---|---|---|---|---|
| 1s8z | 1 Stalkers, 8 Zealots | 1 Stalkers, 8 Zealots | Symmetric & Heterogeneous | Easy |
| 1s9z | 1 Stalkers, 9 Zealots | 1 Stalkers, 9 Zealots | Symmetric & Heterogeneous | Easy |
| **2s3z** | 2 Stalkers, 3 Zealots | 2 Stalkers, 3 Zealots | Symmetric & Heterogeneous | Easy |
| 2s8z | 2 Stalkers, 8 Zealots | 2 Stalkers, 8 Zealots | Symmetric & Heterogeneous | Easy |
| 2s9z | 2 Stalkers, 9 Zealots | 2 Stalkers, 9 Zealots | Symmetric & Heterogeneous | Easy |
| **3s5z** | 3 Stalkers, 5 Zealots | 3 Stalkers, 5 Zealots | Symmetric & Heterogeneous | Easy |
| **3s5z_vs_3s6z** | 3 Stalkers, 5 Zealots | 3 Stalkers, 6 Zealots | Symmetric & Heterogeneous | Super Hard |
| 7s3z | 7 Stalkers, 3 Zealots | 7 Stalkers, 3 Zealots | Symmetric & Heterogeneous | Easy |

hyperparameters settings. We describe our network structure in Tab. 12. This network architecture is used for all experiments in the paper.

## E   EXPERIMENTAL DETAILS

Our experiments were performed on 2 NVIDIA GTX 2080 Ti GPUs. For all the performance curves in our paper, we pause training every 10K timesteps and evaluate for 32 episodes with decentralized greedy action selection. We present the percentage of episodes in which the agents defeat all enemies within the time limit. We now provide details about each part of our experiments.

Table 9: Settings of tasks in the M series. The bolded names indicate the source tasks.

| Map Name | Ally Units | Enemy Units | Type | Difficulty |
|---|---|---|---|---|
| 3m | 3 Marines | 5 Marines | Symmetric & Homogeneous | Easy |
| 4m | 4 Marines | 5 Marines | Symmetric & Homogeneous | Easy |
| 4m_vs_5m | 4 Marines | 5 Marines | Asymmetric & Homogeneous | Hard |
| **5m** | 5 Marines | 5 Marines | Symmetric & Homogeneous | Easy |
| **5m_vs_6m** | 5 Marines | 6 Marines | Asymmetric & Homogeneous | Hard |
| 6m | 6 Marines | 6 Marines | Symmetric & Homogeneous | Easy |
| 6m_vs_7m | 6 Marines | 7 Marines | Asymmetric & Homogeneous | Hard |
| 7m | 7 Marines | 7 Marines | Symmetric & Homogeneous | Easy |
| 7m_vs_8m | 7 Marines | 8 Marines | Asymmetric & Homogeneous | Hard |
| 8m | 8 Marines | 8 Marines | Symmetric & Homogeneous | Easy |
| **8m_vs_9m** | 8 Marines | 9 Marines | Asymmetric & Homogeneous | Easy |
| 9m | 9 Marines | 9 Marines | Symmetric & Homogeneous | Easy |
| 9m_vs_10m | 9 Marines | 10 Marines | Asymmetric & Homogeneous | Easy |
| 10m | 10 Marines | 10 Marines | Symmetric & Homogeneous | Easy |
| **10m_vs_11m** | 10 Marines | 11 Marines | Asymmetric & Homogeneous | Easy |
| 10m_vs_12m | 10 Marines | 12 Marines | Asymmetric & Homogeneous | Super Hard |

Table 10: Settings of the `Spread` tasks. The bolded identities indicate the source tasks.

| Task Identity | # of Agents | # of Landmarks | Field Size | Sight Range | Reach Range |
|---|---|---|---|---|---|
| **2** | 2 | 2 | 6 | 5 | 2 |
| 3 | 3 | 3 | 8 | 7 | 2 |
| **4** | 4 | 4 | 10 | 9 | 2 |
| 5 | 5 | 5 | 10 | 9 | 2 |
| 6 | 6 | 6 | 15 | 14 | 2 |
| **7** | 7 | 7 | 15 | 14 | 2 |
| 8 | 8 | 8 | 15 | 14 | 2 |
| **9** | 9 | 9 | 15 | 15 | 2 |

**Generalization to unseen tasks**  For baselines and ablations, we carried out experiments with 5 different random seeds. In each experiment, we evaluate the trained model for 32 episodes on each unseen task. The results recorded in Tab. 1∼3 are the mean and variance of these 5 random seeds.

**Fine-tuning**  For the performance of fine-tuning MATTAR, we trained 2 source models with different random seeds for each unseen map and used 2 random seeds for each source model for fine-tuning. For learning from scratch, we carried out experiments with 4 different random seeds for each map.

**Multi-task learning**  We carried out experiments with 5 different random seeds for both multi-task learning and learning on a single task. For the experiments of multi-task learning on three tasks shown in the paper, the training sets are {5m, 5m_vs_6m, 8m_vs_9m, 10m_vs_11m}, {2s3z, 3s5z, 3s5z_vs_3s6z}, and {MMM, MMM2, MMM4}, respectively.

**Single-task learning**  For this experiment, we tested each baseline and ablation algorithm with 5 random seeds.

## F  FORWARD MODEL FOR TASK REPRESENTATION LEARNING

In our method, we utilize dynamics modelling to learn task representations which can capture the similarity between different tasks. We use a hypernetwork as the representation explainer to generate the parameters of the forward model. In practical implementation, the forward model consists two components, an encoder and a decoder (Fig. 8(a)).

For the encoder network, we first use the population-invariant embedding layer to get a fixed-dimensional embedding vector and feed it into a fully-connected layer whose parameters are generated

Table 11: Settings of the `Gather` tasks. The bolded identities indicate the source tasks.

| Task Identity | # of Agents | # of Landmarks | Field Size | Sight Range | Reach Range |
|---|---|---|---|---|---|
| **2** | 2 | 1 | 6 | 5 | 2 |
| 3 | 3 | 1 | 8 | 7 | 2 |
| **4** | 4 | 1 | 10 | 9 | 2 |
| 5 | 5 | 1 | 10 | 9 | 2 |
| **7** | 7 | 1 | 15 | 14 | 2 |
| **9** | 9 | 1 | 15 | 15 | 2 |
| 10 | 10 | 1 | 20 | 19 | 2 |
| 15 | 15 | 1 | 20 | 19 | 2 |

Table 12: Hyperparameters about the network structure in our experiments.

| name | value |
|---|---|
| The hidden dimension for mixing network | 32 |
| The number of layers for the hypernet in mixing network | 2 |
| The hidden dimension for the hypernet | 64 |
| The length of the encoding vector of agent ID | 4 |
| The dimension of task representations | 32 |
| The output dimension of the encoder in the forward model | 32 |
| The output dimension of the attention module | 64 |
| The hidden dimension for the query and key in attention module | 8 |

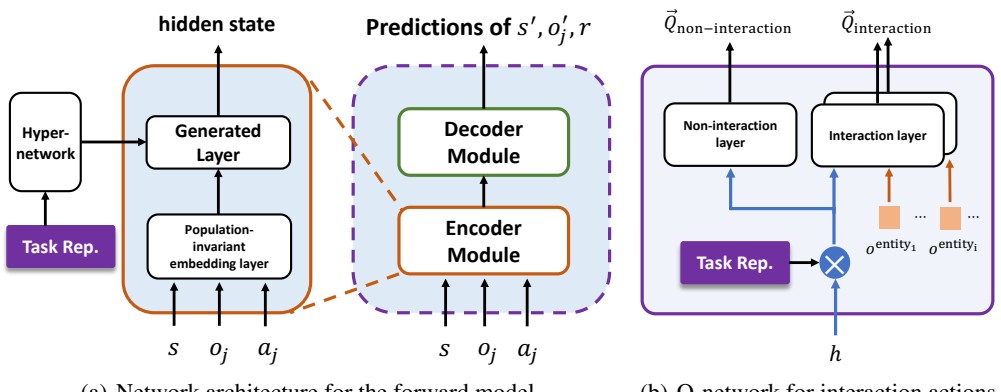

(a) Network architecture for the forward model  (b) Q-network for interaction actions

Figure 8: The architecture of our forward model and the Q-network for interaction actions. In (b), $o^{\text{entity}_i}$ denotes the observation component corresponding to the influence of the $i$-th interactive action, $\vec{Q}_{\text{non-interaction}}$ denotes the Q-values of non-interactive actions, and $\vec{Q}_{\text{interaction}}$ denotes the Q-values of interactive actions.

by the representation explainer. The output hidden variables are fed into the decoder to predict the next state, the next observation, and the global reward. The encoder module and the representation explainer are shared among tasks and are fixed when learning representations for unseen tasks. The decoder module is task-specific, and we allow the decoder to be optimized together with task representations when adapting to unseen tasks.

For the population-invariant embedding layer in the encoder module, like in the policy, we decompose the input state and observation into several entity-specific components, pass them through an embedding layer, and do a pooling operation for output vectors. We also deal with the case of the varying number of actions in the input by incorporating actions into to observation $o_i$. We concatenate non-interaction actions with agent $i$'s own observation component $o_i^{\text{own}}$ and interaction actions with

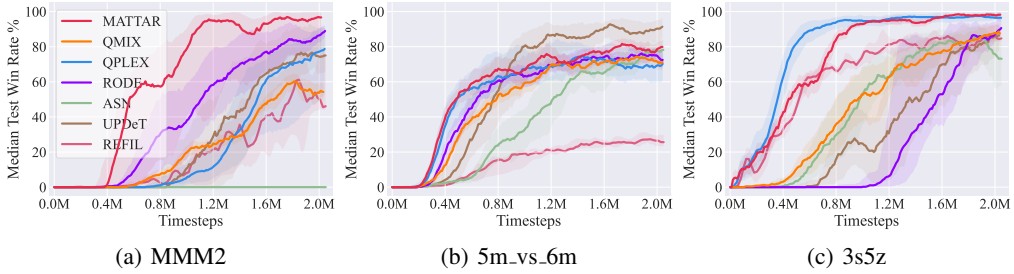

Figure 9: A bonus: when learning from scratch on single tasks, MATTAR architecture exhibits good performance. For performance on more SMAC benchmark, please refer to Fig. 10.

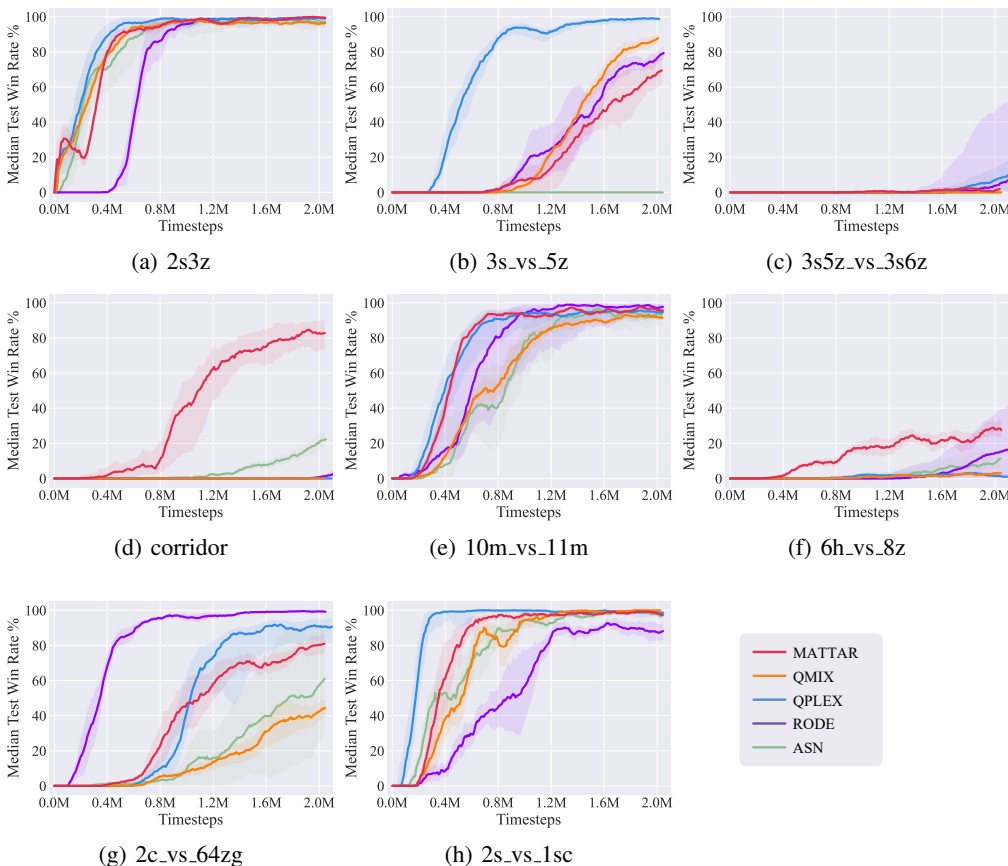

Figure 10: More results for the performance of MATTAR on the SMAC benchmark when learning from scratch on single tasks.

the observation components corresponding to each entity. We also note that other population-invariant structures can also be applied to our approach.

## G  VARYING NUMBERS OF ACTIONS

In some multi-agent environments, there are interaction actions that have semantics relating to other opponents in the environment. In this case, the action dimension is related to the number of opponents, preventing flexible transfer to unseen tasks. To deal with this problem, we adopt the structure shown in Fig. 8(b) for the estimation of Q-values for these interaction actions. In this structure, Q-values

for interaction and non-interaction actions are estimated separately. For non-interaction actions, we use a fully-connected network whose input is the concatenation of observation encoding $h$ and task representation $z$. For an interaction action, we use a network that takes as input the concatenation of $h$, task representation $z$, and the observation component relating to the corresponding opponent.

## H    BONUS: PERFORMANCE ON SINGLE-TASK TRAINING

Although not designed for this goal, we find that MATTAR can outperform state-of-the-art MARL algorithms when trained on some single tasks. Specifically, we remove the task representation module and train MATTAR from scratch. We compare our method with two state-of-the-art value-based MARL baselines (QMIX (Rashid et al., 2018) and QPLEX (Wang et al., 2021a)), a role-based learning algorithm (RODE) (Wang et al., 2021b), and one underlying algorithm of MATTAR which considers the Q-values of interaction actions separately (ASN) (Wang et al., 2020b). For the representative tasks of the three series in the main text (MMM2, 5m_vs_6m and 3s5z), we additionally compare with two methods with similar attentional mechanisms (UPDeT (Hu et al., 2021) and REFIL (Iqbal et al., 2021)).

Fig. 9 shows the learning curves of different methods. We find that our population-invariant network structure achieves comparable performance in all tasks. It is worth noting that this structure even significantly outperforms all other algorithms on the super hard map `MMM2`. In Fig. 10, we show the comparison on more SMAC maps, on which MATTAR also has comparable performance. Given that our underlying algorithm is QMIX, this is an inspiring result. We hypothesize that our self-attention scheme increases the representational capacity by learning to attend to appropriate entities in the environment.

## I    TESTING OF ONE ALTERNATIVE FOR SOURCE TASK DEFINITION

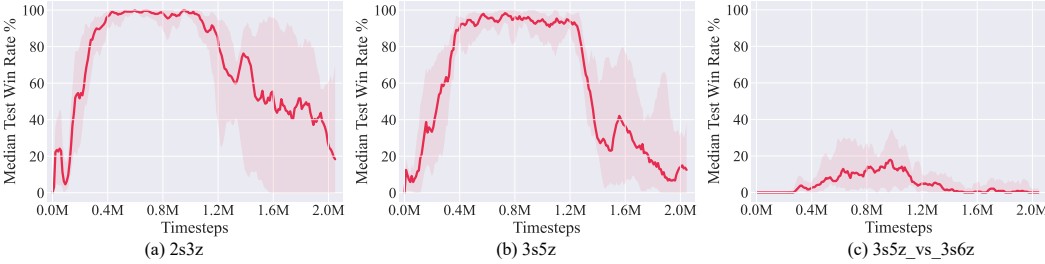

(a) 2s3z                                    (b) 3s5z                                    (c) 3s5z_vs_3s6z

Figure 11: Testing of one alternative for source task definition. In this experiment, we apply a normalization layer on top of the joint learned vectors with the representation explainer, and use it as the task representations for source tasks.

One possible method for determining the source task representations is to joint learn representation explainer and task representations together. However, this practice often brings representations with very small norms. A remedy for this problem is to additionally apply a normalization layer on top of the jointly learned vectors. We conduct experiments to validate this approach, and the results of learning performance on source tasks for the SZ series are shown in Fig. 11. It is interesting that the learning curves on source tasks begin to drop after about 1M samples for all these three tasks. These results show that it is hard to get a meaningful representation space when learning together with the representation explainer. We hypothesize the reason behind the phenomenon is that there are limited signals that can guarantee information about task relationship is encoded in the representation space.

## J    DISCUSSIONS WHEN TASKS UNDER GREAT DIFFERENCES

Our approach shows great advantages over ablations and baselines in the experimental results reported in the main text. To further explore the ability of our approach, we additionally conduct two experiments, and the results are reported in Tab. 13.

Table 13: Two additional experiments where the unseen tasks are quite different.

| | Source Tasks | | | Unseen Tasks | | | |
|---|---|---|---|---|---|---|---|
| | 1s2z | 1s3z | 2s3z | 3s5z | 3s5z_3s6z | 4s7z | 4s7z_4s8z |
| MATTAR | 0.94±0.04 | 0.97±0.03 | 0.91±0.07 | 0.68±0.12 | 0.01±0.01 | 0.39±0.19 | 0.00±0.00 |
| | MMM | MMM2 | MMM4 | 1s8z | 2s3z | 3s5z | 7s3z |
| MATTAR | 0.99±0.01 | 0.85±0.01 | 0.89±0.03 | 0.00±0.00 | 0.00±0.00 | 0.00±0.00 | 0.05±0.07 |

In the first experiment, we design source tasks and target tasks with a huge gap in terms of the number of agents. Specifically, We train MATTAR on three source tasks: 1s2z, 1s3z, and 2s3z, each of which contains a small number of agents and has a difficulty level of "easy", and we test it on tasks with more agents and higher difficulty level. As we can see, the transfer performance drops when the number of agents has a huge increase, and the win rate is 0 on 4s7z_4s8z.

In the second experiment, we train MATTAR on three tasks from the MMM series and test it on some tasks of the SZ series. We can see that when the source tasks are not diverse enough to cover unseen tasks, the transfer performance is close to 0.

From the results, we can see that the transfer performance faces a relatively large drop when our approach transfers from tasks with few agents to those with more agents, and a failure will appear when testing on those hard tasks as the decision skills required in these tasks cannot be acquired by training on those easy tasks. Besides, when we try to transfer across different series of tasks, our approach struggles even when testing on the 2s3z task, which is a quite easy task. This phenomenon indicates that policy transfer across quite different tasks is still a quite hard problem that is worth exploring, and our approach may struggle when the target task can not be well covered by the source tasks. How to overcome this challenge and achieve more general policy transfer in multi-agent reinforcement learning remains an open problem.

