# OpenReview forum: "Multi-Agent Policy Transfer via Task Relationship Modeling"
_ICLR.cc/2023/Conference — Submitted to ICLR 2023_

### Official Review · Reviewer_oPef · 2022-10-21

**Confidence:** 4
**Correctness:** 2
**Technical Novelty And Significance:** 3
**Empirical Novelty And Significance:** Not applicable
**Recommendation:** 6

**Clarity, Quality, Novelty And Reproducibility:**

Method
* The notion of "task" is not well-defined in section 2. Source and target tasks in the experiments seem to only differ in the population size. The environment dynamic (for each agent) and the reward function remain the same.
* Why does backprop through the representation result in small-norm vectors?

Experiments
* Can you explain why the performance of MATTAR on MPE is not very different from "0 task rep" and "w/o task rep"?
* Why does performance of MATTAR drop significantly on SMAC **source** tasks when zero-ing out task representation? Why does this phenomenon not happen on MPE?
* Can you show the accuracy of the learned forward model? How good is it?

**Strength And Weaknesses:**

**Strengths**
* The paper is clearly written. The method is well-described. Experiment hypotheses are thoroughly constructed.
* To my best knowledge, the method seems to be novel, although I am not too familiar with the multi-agent RL literature. It is also technically sound.

**Weaknesses**

Method
* The major weakness is that learning a new task representation requires collecting diverse trajectories on the new task. The paper uses the policy trained on source tasks to collect trajectories on a new task. The policy thus needs to generalize decently to the unseen tasks to generate interesting trajectories. I think this point needs to be emphasized in the paper.
* Initializing the representations of the source tasks as orthogonal vectors ignores the relationships between them.
* The learned forward model may be used for planning but only the task representation is used.
* This is not really a weakness. But the method seems to solve more general problem than generalizing to new population size. Based on the current formulation, I believe it can also be applied to a single-agent setting. Is there any particular reason for formulating and showcasing it on an multi-agent setting?

Experiments
* Results on the MPE tasks are not convincing. The task representation makes very little difference (first 3 rows of tables 4 and 5). The authors incorrectly claim "MATTAR...  significantly outperforms the baselines and ablations".

**Summary Of The Paper:**

The paper proposes a method for policy transfer in a multi-agent reinforcement learning setting. The method focuses on transferring to new population size. It learns task representations that capture relationships among source tasks. On an unseen task, a new representation is learned as a linear combination of the learned representations. The unseen task's representation is then plugged into the Q-learning population-invariant policy.

The method is evaluated on two multi-agent benchmarks. It outperforms baseline methods on unseen tasks. The task representation enables higher performance after fine-tuning the policy. It also effectively leverages the multi-task learning setting.

**Summary Of The Review:**

Overall, I like the proposed method as it is very general. But some of results are not convincing and the authors make false claims instead of explaining the observed phenomenon. The paper is borderline and I am learning towards rejection.

=======After rebuttal=======

I thank the authors for giving very detailed and comprehensive responses. I decided to raise the score to 6, hoping that the authors will incorporate their responses into the final version, especially the explanation about the ineffectiveness of the learned task representation on MPE.

---

> ### Author Response · Authors · 2022-11-19
> **Answers to questions (Part 3/3)**
>
> **Q7**: Why does backprop through the representation result in small-norm vectors?
>
> **A7**: We observed this phenomenon in our experiments, and we conjecture that it is an optimization issue. We feel sorry that we cannot give a clearer explanation at this stage.
>
> **Q8**: Concerns about the experimental results on MPE benchmarks.
>
> **A8**: We appreciate your thoughtful concerns about our experimental results. Our opinion is that **unlike SMAC**, MPE tasks have stronger similarities between tasks with different population sizes, especially for the Gather task. The reason is that the agents are always expected to reach the goal position on MPE tasks and full-observability is applied which means that the agents can observe the goal. Thus, the behavior patterns for different tasks are relatively homogeneous. This may weaken the need for task differentiation and reduce the importance of task representation. However, in SMAC, the neural network needs to rely on task representation to distinguish between different tasks, which means that the task representation input could have a large impact on the neural network's output. This is the reason why the ablation "0 task rep." got a very terrible performance as the zero vector deviates significantly from the representation inputs the policy has seen in the training phase. While on MPE tasks, the phenomenon is alleviated due to the nature of the MPE tasks, and this is also the reason why the performance of MATTAR on MPE is not significantly different from "0 task rep." and "w/o task rep.". However, we still observe that MATTAR exhibits overall advantages compared to ablations and baselines in such a benchmark which is likely different from SMAC.
>
> **Q9**: Can you show the accuracy of the learned forward model? How good is it?
>
> **A9**: We show the prediction loss of the learned forward model in the *sz* series of the SMAC benchmark in the table below. Specifically, we show the prediction loss on three source tasks, 2s3z, 3s5z, 3s5z\_vs\_3s6z, and one target task 3s5z\_vs\_3s7z. The prediction loss for state and observation are $L_2$ distance between the predicted vector and the true vector, while the prediction loss for reward is defined as the squared error. We also give the statistics of the true state, observation, and reward for reference. Specifically, we studied the $L_2$ norm of state and observation, as well as the squared value of the reward. Their mean and standard deviation are respectively $6.6767\pm0.7368,4.5777\pm3.5165,0.0175\pm0.0368$. Thus, the results can validate the effectiveness of our forward model learning process.
>
> | Prediction Loss | 2s3z                  | 3s5z                  | 3s5z\_vs\_3s6z        | 3s5z\_vs\_3s7z        |
> |-----------------|-----------------------|-----------------------|-----------------------|-----------------------|
> | State           | 0.509$\pm$0.014       | 0.4$\pm$0.004         | 0.405$\pm$0.01        | 0.402$\pm$0.005       |
> | Observation     | 0.311$\pm$0.014       | 0.25$\pm$0.009        | 0.264$\pm$0.01        | 0.264$\pm$0.004       |
> | Reward          | 2.651e-3$\pm$0.297e-3 | 1.159e-3$\pm$0.123e-3 | 1.147e-3$\pm$0.216e-3 | 1.016e-3$\pm$0.022e-3 |
>
> Ref:
>
> [1] Camacho, Eduardo F., and Carlos Bordons Alba. Model predictive control. Springer, 2013.
>
> [2] Janner, Michael, et al. When to trust your model: Model-based policy optimization. NeurIPS 2019.
>
> [3] Rakelly, Kate, et al. Efficient off-policy meta-reinforcement learning via probabilistic context variables. ICML 2019.
>
> [4] Zhang, Jin, et al. Metacure: Meta reinforcement learning with empowerment-driven exploration. ICML 2021.
>
> [5] Zhu, Zhuangdi, Kaixiang Lin, and Jiayu Zhou. Transfer Learning in Deep Reinforcement Learning: A Survey, arXiv preprint arXiv:2009.07888.

---

> > ### Comment · Reviewer_oPef · 2022-11-23
> > **Thank you for your replies!**
> >
> > Thank you very much for the thorough replies. I have read and don't have any more questions.

---

> > > ### Author Response · Authors · 2022-12-05
> > > **Thank you very much!**
> > >
> > > We really appreciate the reviewer's comments and suggestions. We will take your suggestion and pro-actively clarify these questions in our final version. Thanks very much for your raising the score!

---

> ### Author Response · Authors · 2022-11-19
> **Answers to questions (Part 2/3)**
>
> **Q4**: Based on the current formulation, I believe it can also be applied to a single-agent setting. Is there any particular reason for formulating and showcasing it on an multi-agent setting?
>
> **A4**: The primary motivation of our work is to achieve team adaptation in the multi-agent setting, and our approach is proposed to solve this problem. Actually, our approach considers the multi-agent setting as it handles the varying input lengths and considers the agent interaction for task representation learning. We do find that our method can fit the single-agent setting with slight modification, and we expect to investigate its effectiveness in single-agent tasks in the future. Thanks for the constructive suggestion!
>
> **Q5**: Results on the MPE tasks are not convincing. The task representation makes very little difference (first 3 rows of tables 4 and 5). The authors incorrectly claim "MATTAR ... significantly outperforms the baselines and ablations".
>
> **A5**: This is indeed a misuse, and we have modified our statement in the revision. However, we still notice that MATTAR has an overall advantage on the Spread task compared to its two ablations, with 45\% higher than w/o task rep. and 16\% higher than 0 task rep.
>
> **Q6**: The notion of "task" is not well-defined in section 2. Source and target tasks in the experiments seem to only differ in the population size. The environment dynamic (for each agent) and the reward function remain the same.
>
> **A6**: Informally, "task" in the multi-agent setting is a tuple <the transition function, the reward function, the number of agents>. We do not assume specific task structures in section 2. The rationale is that the transition and reward functions generally depend on the joint action and global state, so the two functions naturally change with the number of agents. And the tasks become different when the transition functions, reward functions, or the number of agents are different. Actually, we note that even when only the population size varies, the strategy of each agent to solve the task can be totally different, because the results of the agent policy rely on the other agents, and different agent interaction patterns are required when the population size changes, which implies the environment dynamics and the reward function for each agent also change. For example, on SMAC, "3s5z vs 3s6z" (3s5z\_3s6z) is much harder than "3s5z vs 3s5z" (3s5z). Besides, we also note that dynamic population size is a common situation for team adaptation in reality, which inspires us to design the experiments.

---

> ### Author Response · Authors · 2022-11-19
> **We really appreciate your detailed reviews and interest in our paper! We provide some explanations for your questions here and we hope they can be helpful for you. (Part 1/3)**
>
> **Q1**: The major weakness is that learning a new task representation requires collecting diverse trajectories on the new task. The paper uses the policy trained on source tasks to collect trajectories on a new task. The policy thus needs to generalize decently to the unseen tasks to generate interesting trajectories.
>
> **A1**: Firstly, we apologize that we missed an important detail that both in the training and transfer phase, the data utilized to train the task representation explainer or task representations are collected with **random policy**, and we have **emphasized this in our revision**. Actually, we can not directly apply the trained policy to collect data because it needs to insert the target task representation which is not available at the beginning. We agree that utilizing random policy is relatively crude and it may fail in some scenarios requiring long-horizon exploration. Currently, we find it works well in scenarios with dense agent interactions and dense rewards, e.g. SMAC. For more complex scenarios, we believe training a specific probing policy for high-efficient task representation learning is a good idea, however, it is not the current focus of this paper.
>
> **Q2**: Initializing the representations of the source tasks as orthogonal vectors ignores the relationships between them.
>
> **A2**: Our approach is based on the assumption that we do not have any prior knowledge about the relationships of source tasks because in many scenarios it is difficult to quantify the similarity between two tasks in advance. Such as in SMAC, it is hard to say whether 3s5z is more similar to 2s4z or 3s6z. In such a way, we apply a general practice that defines the source task representations as orthogonal vectors, thus avoiding making potentially incorrect assumptions about the relationships between tasks. In fact, we capture the relationships between tasks via training a shared representation explainer and policy network, as representations of similar tasks will be interpreted similarly. To further justify that our practice is reasonable, we additionally conduct an experiment where we compare with a baseline that employs random initialized source task representations, and the results are listed as follows (ten random seeds are conducted for this supplemental experiment).
>
> |                 | Source Task 1          | Source Task 2          | Source Task 3          | Unseen Task 1          | Unseen Task 2          | Unseen Task 3          | Unseen Task 4          | Unseen Task 5          |
> | --------------- | ---------------------- | ---------------------- | ---------------------- | ---------------------- | ---------------------- | ---------------------- | ---------------------- | ---------------------- |
> |                 | 2s3z                   | 3s5z                   | 3s5z\_3s6z             | 1s8z                   | 1s9z                   | 2s8z                   | 2s9z                   | 7s3z                   |
> | MATTAR          | **0.99**$\pm$0.01 | **0.99**$\pm$0.02 | **0.44**$\pm$0.18 | **0.80**$\pm$0.08 | **0.65**$\pm$0.15 | **0.90**$\pm$0.09 | **0.87**$\pm$0.06 | **0.16**$\pm$0.12 |
> | random src rep. | 0.98$\pm$0.03          | 0.97$\pm$0.04          | 0.43$\pm$0.25          | 0.66$\pm$0.22          | 0.55$\pm$0.19          | 0.89$\pm$0.09          | 0.86$\pm$0.10          | 0.03$\pm$0.02          |
>
> **Q3**: The learned forward model may be used for planning but only the task representation is used.
>
> **A3**: There are indeed a series of model-based reinforcement learning (MBRL) algorithms utilizing the learned model to do planning [1] or generate more training data [2]. However, the purpose of our approach is to capture the feature of the task via the model learning process instead of learning an accurate model. Similar ideas could be found in some previous works [3][4] and planning has hardly been studied in transfer learning[5]. Besides, planning in the multi-agent setting is more difficult than in the single-agent setting, since the joint action space can be much larger. With these concerns, we do not conduct planning now. We will investigate the performance of planning in the future.

---

### Official Review · Reviewer_AfTn · 2022-10-25

**Confidence:** 2
**Clarity, Quality, Novelty And Reproducibility:** Please see above for detailed comments.
**Correctness:** 3
**Technical Novelty And Significance:** 3
**Empirical Novelty And Significance:** 3
**Recommendation:** 6

**Strength And Weaknesses:**

Strength
- The proposed framework seems novel and reasonable to the problem considered.
- Nice presentation with many useful figures and intuitive discussions so that the paper is easy to follow.
- Promising experimental results.

Weaknesses/Questions
- Can you provide some theoretical characterization of the proposed framework? Transfer learning for multi-agent setting is very complicated and challenging, due to varying population size, input dimension, and even different action spaces. For example suppose source domain action space is a continuous interval [0, 1] while target domain action space is finite {0, 1}, then it is guaranteed that the two tasks are not transferable in some scenarios. For example, in single-agent case, which is simpler than multi-agent, [1] studies the alignment of two MDPs.
- When do you expect the transfer to unseen tasks will succeed or fail? How many samples are needed to reliably learn a task representation for unseen task. It seems to achieve this kind of generalization across tasks, you may need datasets from a lot of different tasks, since the latent space for task representation is quite large and many values can be undefined.
- Does the proposed framework also apply to competitive/mixed multi-agent system?

[1] Domain Adaptive Imitation Learning

**Summary Of The Paper:**

The paper studies transfer learning in multi-agent setting, where existing works based on graph neural networks or attention mechanisms achieves generalization across tasks implicitly through the generalization of neural network function approximations. The authors propose to explicitly model the task relationships by learning a task embedding and the resulting policies will depend on the task representations, which can be inferred for new tasks given a few trajectories. Experimental results demonstrates the effectiveness of the proposed framework.

**Summary Of The Review:**

Overall I think this is a well-written paper with a reasonable framework to capture the task relationships for multi-agent transfer learning. The paper could be improved with some theoretical characterizations of the problem and the proposed framework.

---

> ### Author Response · Authors · 2022-11-19
> **Sincerely thank you for your constructive reviews! We offer some clarification to your questions here, and we look forward to your further comments!**
>
> **Question 1**: Can you provide some theoretical characterization of the proposed framework?
>
> **Answer 1**: This is indeed an interesting topic. Actually, as you point out that some tasks are not transferable in some scenarios, such as in your proposed example where the action spaces of two tasks are hugely different. In fact, currently, we are investigating whether we can theoretically offer some guarantees for the advantages of our task representation mechanism in our problem setting. In this process, we need certain assumptions about the source and target tasks, and the concept of MDP alignment you mentioned is quite inspiring. We believe the study about MDP alignment in the multi-agent setting and more theoretical guarantees of our approach are valuable future work directions.
>
> **Question 2**: When do you expect the transfer to unseen tasks will succeed or fail? How many samples are needed to reliably learn a task representation for unseen task.
>
> **Answer 2**:
>
> - From our point of view, our approach works when the source tasks can provide some policy knowledge for accomplishing the target task. This requires there exist certain similarities between the cooperation patterns of the source tasks and the target task. For example, some skills like firesetting and retreating in SMAC benchmark are reusable between different tasks. In fact, we have provided some discussions about the cases where policy transfer between tasks may be super hard and we also pasted the results below. In our provided experiments, our approach struggles when there exists a huge change in agent numbers between source and target tasks or the map types are completely different. This study can provide us with some inspiration and side-by-side validate our hypothesis.
> |        | Source Task 1 | Source Task 2 | Source Task 3 | Unseen Task 1 | Unseen Task 2 | Unseen Task 3 | Unseen Task 4 |
> | ------ | ------------- | ------------- | ------------- | ------------- | ------------- | ------------- | ------------- |
> | Exp. 1 | 1s2z          | 1s3z          | 2s3z          | 3s5z          | 3s5z\_3s6z    | 4s7z          | 4s7z\_4s8z    |
> | MATTAR | 0.94$\pm$0.04 | 0.97$\pm$0.03 | 0.91$\pm$0.07 | 0.68$\pm$0.12 | 0.01$\pm$0.01 | 0.39$\pm$0.19 | 0.00$\pm$0.00 |
> | Exp. 2 | MMM           | MMM2          | MMM4          | 1s8z          | 2s3z          | 3s5z          | 7s3z          |
> | MATTAR | 0.99$\pm$0.01 | 0.85$\pm$0.01 | 0.89$\pm$0.03 | 0.00$\pm$0.00 | 0.00$\pm$0.00 | 0.00$\pm$0.00 | 0.05$\pm$0.07 |
>
> - In terms of the samples needed to learn a task representation for the unseen task, in practice, we uniformly utilize $50K$ (transition) samples to learn the task representation, and we find it is enough for the prediction loss to converge stably.
>
> **Question 3**: Does the proposed framework also apply to competitive/mixed multi-agent system?
>
> **Answer 3**: We mainly focus on the transfer of cooperation in this paper and our experiments are all conducted in cooperative tasks. An extension of our framework to competitive/mixed multi-agent systems and investigating the similarities of Nash/correlated equilibrium between different tasks for transfer is an interesting topic. However, currently we have no particularly clear conclusions.

---

> ### Author Response · Authors · 2022-12-05
> **Dear Reviewer AfTn, have our responses addressed your questions?**
>
> We sincerely thank you again for your valuable reviews and hope our responses could address your questions. As the response system will end in one week, please kindly let us know if we missed anything. More questions on our paper are always welcomed. Thanks a lot again!
>
> Sincerely yours,
>
> Authors of Paper1950

---

### Official Review · Reviewer_dhhQ · 2022-11-02

**Confidence:** 4
**Correctness:** 2
**Technical Novelty And Significance:** 2
**Empirical Novelty And Significance:** 1
**Recommendation:** 3

**Clarity, Quality, Novelty And Reproducibility:**

The clarity and reproducibility are excellent. The quality of the algorithm is good, but the quality of the experimental design is lacking, as explained in Strengths and Weaknesses (S&W). There does not seem to be much novelty in the algorithm or in the experimental design, which is also explained in S&W.

**Strength And Weaknesses:**

Strengths
The paper motivates and addresses an important AI problem: fully-cooperative multi-agent learning, which can be and has often been denoted as just multi-task learning for several years (Parisotto et al., 2015).

MATTAR is simple, easy-to-implement, and parametrizes a model that simultaneously outputs the next state, next observation, and global reward. This algorithm can handle varying input and output sizes, which is often the case across differing tasks, using a population-invariant network (PIN). However, the use of a high-level model that maps task representations to model parameters is not new nor is the use of PINs new, as the paper itself claims on page 4. In fact, it is unclear what the paper's contribution regarding PINs is, so the paper would benefit by clearly explaining and justifying this. Perhaps this is in Appendix G, but regardless, I apologize if this is clearly stated in the paper and I missed it.

The paper graciously provides a case study on a toy problem to help readers gain an intuitive understanding of how MATTAR works, but this case study doesn't include visualization of trajectories to task 3, even though it corresponds to one of the two non-zero learned coefficient components, and it omits the fact that from Figure 3(b), it is also typical behavior in task 4 for the agent to move left when justifying why the agent moves up first in moving towards the unseen task goal. I assume that the agent moves up instead of left first actually because unseen goal is further up than left from the initial agent region or because the trajectory from only one rollout or one policy is being shown.

The paper is easy to read, clear, and the Figures, Tables, etc. are great at capturing and explaining salient points the paper makes.  It does a great job of providing enough information on the experimental design to reproduce the experiments.

The paper evaluates MATTAR on multi-task settings on two benchmarks, SMAC and MPE, which have varying degrees of difficulty. Experiments are carried out with five random seeds, which in some cases is sufficient, but in certain cases here is insufficient. The paper claims that "... the super hard map 3s5z_vs_3s6z and [sic] cannot be solved by learning from scratch," which is too strong of a claim generally, but especially if only 5 seeds were run.

The paper shows that MATTAR generally outperforms multiple baselines on multiple multi-task transfer learning setups. However, I'm unconvinced that the baselines are competitive, as UPDeT doesn't even aim to address the same problem setting as MATTAR does (single source task vs. multi-source task transfer) though the paper does aim to provide a fair comparison by introducing UPDeT-b. Regardless, there exist a plethora of multi-source task transfer learning algorithms (often i.e., fully-cooperative multi-agent learning algorithms) that may be better suited to compare against. The paper would also benefit by evaluating MATTAR on a set of more diverse benchmarks than just closed, game or toy-type settings, such as Starcraft (SMAC) and MPE, especially one or two real-world settings so that the research community has a better idea of how MATTAR performs generally.

---

Parisotto, Emilio, Jimmy Lei Ba, and Ruslan Salakhutdinov. "Actor-mimic: Deep multitask and transfer reinforcement learning." arXiv preprint arXiv:1511.06342 (2015).

**Summary Of The Paper:**

This paper puts forth a fully-cooperative multi-agent/task algorithm, MATTAR, that can be used to transfer knowledge learned from source tasks to improve learning unseen tasks at test time. MATTAR learns a high-level mapping from task representations to state, observation, and reward function parameters and combines this with a policy learning approach that allows for inputs and outputs of varying sizes. The paper evaluates MATTAR on multiple benchmarks against multiple baselines and shows that MATTAR generally attains superior performance in terms of win rate, sample efficiency, and feasibility (i.e., winning at all).

**Summary Of The Review:**

The paper clearly motivates and addresses an important AI problem. It proposes the algorithm MATTAR and shows that it generally outperforms what it believes to be competitive baselines on two game/simulation benchmarks extended to multi-task transfer learning settings.

However, I believe that this paper falls far short of acceptance due to the lack of novelty, limited experimentation, poor experimental design, and lack of analysis of results.

---

> ### Author Response · Authors · 2022-11-19
> **Clarification for the concerns (Part 2/2)**
>
> **Weakness 3**: Five random seeds are not sufficient for some claims.
>
> **Answer 3**: Five random seeds are widely used and considered sufficient in the MARL setting[1]. However, we agree that the claim "3s5z\_vs\_3s7z cannot be solved by learning from scratch" is too absolute with five random seeds. We have modified the statement in our revision and additionally conducted five random seeds for the fine-tuning experiment on 3s5z\_vs\_3s7z. The average results of ten random seeds are listed in the table below, where we also report the test return values besides the win rate. We find learning from scratch is hard to achieve a positive win rate, while fine-tuning MATTAR can help achieve a higher win rate which implies the effectiveness of the transfer.
>
> | 3s5z\_vs\_3s7z | MATTAR fine-tuning | Learn from scratch (w/ repr.) | Learn from scratch (w/o repr.) | QMIX          | QPLEX          |
> |------------------|--------------------|-------------------------------|--------------------------------|---------------|----------------|
> | Test Win Rate    | 0.21$\pm$0.18      | 0.00$\pm$0.00                   | 0.00$\pm$0.00                    | 0.00$\pm$0.0   | 0.00$\pm$0.00    |
> | Test Return      | 16.69$\pm$1.39     | 11.18$\pm$2.45                | 12.76$\pm$0.78                 | 9.35$\pm$2.99 | 10.52$\pm$0.42 |
>
> **Weakness 4**: I am unconvinced that the baselines are competitive and the paper would benefit by evaluating MATTAR on more diverse benchmarks, especially one or two real-world settings.
>
> **Answer 4**:
>
> - In fact, UPDeT and REFIL are currently state-of-the-art methods concerning transfer learning or multi-task learning in the setting of multi-agent tasks. Common cooperative multi-agent reinforcement learning algorithms fail to fit into our setting as they can not handle varying input lengths. Thus, we select these two algorithms as our main baselines, and we include two ablations to provide a clearer analysis of our approach. Also, for single-task learning shown in Appendix H that does not concern varying input lengths, we have provided more diverse baselines. Indeed, this reveals the fact that there is still a lot of room for development in the field of multi-agent transfer learning, and we believe our work is a good step forward in this area.
>
> - We appreciate your constructive suggestion about the benchmarks. Actually, we focus on team adaptation, or to say, the transfer of cooperation in our paper. StarCraft Multi-Agent Challenge (SMAC) and MultiAgent Particle Environment (MPE) are two of the most familiar benchmarks for evaluating the cooperative performance of multi-agent algorithms, and we choose them as the benchmarks at the current stage. We agree that more evaluations on benchmarks of real-world settings can offer a deeper understanding of our approach, which is a valuable future work direction.
>
> Ref:
>
> [1] Gorsane, Rihab, et al. Towards a Standardised Performance Evaluation Protocol for Cooperative MARL. NeurIPS 2022.

---

> ### Author Response · Authors · 2022-11-19
> **Sincerely thanks for your thoughtful comments! Here we offer some explanations to clarify your concerns and we hope they can be helpful. (Part 1/2)**
>
> **Weakness 1**: The main concern about the novelty of this work.
>
> **Answer 1**: To begin with, we want to emphasize that our work pioneers explicitly modeling task relationships to help policy transfer in the multi-agent setting, and we believe it is a good contribution to the community. Specifically, the contribution of our work mainly consists of two parts: the task representation learning schema and the population-invariant network (PIN).
>
> - For task representation learning, though utilizing a hyper-network to map task representations to model parameters is not new, our mechanism is **quite different** from previous typical practices as (1) it applies a novel two-stage learning scheme and (2) it contemplates the multi-agent setting, allows varying input lengths between tasks and considers agent interaction for task representation learning.
>
> - For **the contribution regarding population-invariant network (PIN)**, we apologize for the unclear statement. Actually, though the usage of PIN is not novel, our PIN is a lightweight design compared with previous methods and we provide a comparison of parameter numbers of different methods in the SMAC task 10m\_vs\_11m in the table below.
> Besides, it works pretty well empirically compared to some other practices. The results in Appendix H demonstrate the superiority of our PIN as the performance gains entirely come from the PIN design, and we believe this can provide some inspirations for related work. We have **stressed this in our revision** to express the contribution regarding PIN more clearly.
> |              | MATTAR | UPDeT  | REFIL  |
> |--------------|--------|--------|--------|
> | 10m\_vs\_11m | 96232  | 420178 | 149991 |
>
> **Weakness 2**: Concerns about the toy study.
>
> **Answer 2**: We only selectively show trajectories for five source tasks in our paper for simplicity. We do not include the trajectories on task 3 because it would result in too many symbols on the top and make it look messy. We study the trajectories for task 3, and we find that the typical behavior for task 3 is also to move up first. In terms of the unseen target task, if we showed more trajectories, we could find some trajectories moving up first and others moving left first. This depends on the initial position of the agent and the specific learned model. However, it is consistent with our claim as the learned representation guides the agent to reuse the strategies (moving up, moving left) on the two most similar source tasks (task 3: moving up; task 4: moving up, moving left) when the agent has just been initialized and the target goal is out of the agent's sight.

---

> ### Author Response · Authors · 2022-12-05
> **Dear Reviewer dhhQ, have our responses addressed your questions?**
>
> We sincerely thank you again for your thoughtful comments and hope our responses could address your questions. As the response system will end soon, please kindly let us know if we missed anything. Further questions on our paper are always welcomed. If there are no more questions, we will appreciate it if you can kindly raise the score.
>
> Sincerely yours,
>
> Authors of Paper1950

---

### Official Review · Reviewer_U9UT · 2022-11-02

**Confidence:** 4
**Correctness:** 4
**Technical Novelty And Significance:** 2
**Empirical Novelty And Significance:** 3
**Recommendation:** 6

**Clarity, Quality, Novelty And Reproducibility:**

**Clarity:** The paper is generally well-written and conveys the main insights well.
**Novelty:** The idea of pre-defining the task representation vectors as mutual orthogonal vectors and learning linear weight vectors is new in the MARL context. However, as mentioned in weakness #1, some ideas overlap with prior works.
**Reproducibility:** The source code is provided in the supplementary material to reproduce the results.

**Strength And Weaknesses:**

**Strength:**
1. The paper is generally well-written and conveys the main methods well.
2. Experiments in Sections 4.1-4.3 show the benefits of MATTAR against competitive baselines. The ablation studies and related empirical analyses improve the understanding of MATTAR.

**Weaknesses:**
1. Novelty can be limited because 1) the idea of learning the small-size task parameters (i.e., the weight vector $\mu$) for adaptation is similar to Zintgraf et al. (2021) and 2) the task policy learning component is based on PIN.
2. I agree that the idea of pre-defining the task representation vectors as mutual orthogonal vectors and learning linear weight vectors is new and interesting. Because this idea is the main contribution, this paper can benefit from having a comparison and discussion (i.e., pros and cons) w.r.t. this idea. For instance, compared to Zintgraf et al. (2021), linear weight learning has advantages, including intuitive understanding and well-formed representation space. However, this linear weight learning can have disadvantages as mentioned in the conclusion (future work). Not necessary, but I wonder whether there can be a small experiment comparing the ideas between the linear weight learning and non-confined learning (similar to Zintgraf et al. (2021)).
3. Related to the weakness #2, if all source tasks are set up to be identical, can the mutual orthogonal vector idea hurt the performance?
4. MATTAR applies fine-tuning for unseen tasks. I wonder it would be more fair for baselines in Table 1-5 to apply fine-tuning.

**Minor:**
1. In Section 2, for completeness, source tasks $\{S_i\}$ and unseen tasks $\{T_j\}$ can be mathematically defined using $G$ (Dec-POMDP) with different transitions and reward functions.

**Summary Of The Paper:**

This paper presents MATTAR, which learns the common structure of tasks and transfers this knowledge to unseen tasks. Specifically, the framework first learns the representation explainer based on the pre-defined and mutual orthogonal task representation vectors. Then, during the transfer phase, a task representation vector for a new task is learned for adaptation. MATTAR also includes the task policy learning component that leverages PIN for addressing the varying number of agents across tasks. Experiments in the SMAC and MPE domains demonstrate the effectiveness of MATTAR's transfer capability.

**Summary Of The Review:**

Overall, I have a positive evaluation of this paper. While the novelty can be a concern, the experimental results are solid and detailed, and the paper conveys the main findings clearly. Therefore, I initially vote for 6 (marginally above the acceptance threshold) and will make a final decision on the recommendation after the authors' response.

---

> ### Author Response · Authors · 2022-11-19
> **Clarifications for the weaknesses (Part 2/2)**
>
> **Weakness 3**: If all source tasks are set up to be identical, can the mutual orthogonal vector idea hurt the performance?
>
> **Answer 3**: We think this violates the basic multi-task setting and is only a worst-case for the proposed method to test its robustness.
> We argue that the mutual orthogonal vector idea itself does not hurt the performance even when some tasks are identical, since it is feasible to train multiple low-dimensional representations for the same single task. We conduct a small experiment where four source tasks are all set up to be 2s3z to verify this point. We report the training performance on source task 2s3z and the generalization performance on two unseen tasks in the table below. Besides, a comparison to ablation "w/o task rep." is provided. We find that our approach performs well even in this worst case, which justifies that the mutual orthogonal vector idea does not hurt the performance.
>
> |               | Source Task   | Unseen Task 1 | Unseen Task 2 |
> |---------------|---------------|---------------|---------------|
> |               | 2s3z          | 1s3z          | 2s2z          |
> | MATTAR        | 1.0$\pm$0.0   | 0.93$\pm$0.08 | 0.73$\pm$0.27 |
> | w/o task rep. | 0.99$\pm$0.01 | 0.84$\pm$0.07 | 0.86$\pm$0.14 |
>
> Moreover, this issue is of less value in practice, as we can easily bypass it by treating these identical tasks as a single task.
>
> **Weakness 4**: MATTAR applies fine-tuning for unseen tasks. I wonder it would be more fair for baselines in Table 1-5 to apply fine-tuning.
>
> **Answer 4**: To begin with, we apologize for the omission of an important detail in our paper that we only utilized the random policy to collect the data for task representation learning in the transfer phase. We have **stressed this detail in our revision**. For experimental results in Table 1-5, all algorithms did not fine-tune the policy network, and the purpose is to compare the generalization performance without policy fine-tuning for different algorithms. Indeed, our approach MATTAR utilized a few random samples to learn the target task representation. Although it takes some extra costs, we argue that it is acceptable, e.g., we could have a small offline dataset for the target task in many scenarios and this allows our representation learning, while the small offline dataset is usually not efficient for fine-tuning the policy.
>
>
> Ref:
>
> [1] Rakelly, Kate, et al. Efficient off-policy meta-reinforcement learning via probabilistic context variables. ICML 2019.
>
> [2] Zhang, Jin, et al. Metacure: Meta reinforcement learning with empowerment-driven exploration. ICML 2021.

---

> > ### Comment · Reviewer_U9UT · 2022-11-22
> > **Response to Rebuttal**
> >
> > Thank you for the response and for adding additional experiments based on my feedback. This response addresses my main concerns. I have also read the other reviewers' feedback and the corresponding authors' responses. I understand the concerns w.r.t. experimental results (dhhQ, oPef) and also partially agree with the associated authors' rebuttal. I maintain my score and look forward to discussing more in detail with reviewers and AC.

---

> > > ### Author Response · Authors · 2022-12-05
> > > **Thanks a lot for your feedback!**
> > >
> > > Thanks a lot for your reading our rebuttal and the timely response. We also thank you for reading our responses to other reviewers' feedback. We are glad that the main concerns have been addressed and please kindly let us know if you had any further questions in the further discussions with other reviews and AC. Thank you very much!

---

> ### Author Response · Authors · 2022-11-19
> **Thanks for your detailed reviews! We offer some clarification to your concerns here, and we would appreciate it if you had any further comments. (Part 1/2)**
>
> **Weakness 1**: Concerns about the limited novelty.
>
> **Answer 1**: The contribution of our work is composed of two parts: (1) the task representation learning mechanism and (2) the population-invariant network (PIN).
>
> - For the first part, we agree that both Zintgraf et al. (2021) and our approach employ the idea of learning the small-size task parameters, and actually learning small-size task representations is not uncommon in the field of meta learning and transfer learning, e.g., PEARL[1], Meta-CURE[2]. This is not the core idea of our approach, and the main differences between our approaches and these previous practices are that: (a) it is designed for multi-agent settings and considers the agent interaction for task representation learning; (b) we apply a novel two-stage learning scheme and the experiments empirically validate the effectiveness of this practice. One thing that deserves note herein is the task representation in our work is fixed in a task during the policy learning and execution, whereas the representation in previous approaches commonly changes with different input observations even if the task is the same.
>
> - In terms of the second part, the advantage of our design is that it is lightweight (below we offer a comparison for network parameter numbers of different methods in the SMAC task 10m\_vs\_11m). Besides, the results of single-task learning in Appendix H prove the superiority of our PIN design as it brings the gain of better cooperation performance.
> |              | MATTAR | UPDeT  | REFIL  |
> |--------------|--------|--------|--------|
> | 10m\_vs\_11m | 96232  | 420178 | 149991 |
>
> **Weakness 2**: Advantages of our task representation learning scheme.
>
> **Answer 2**: The main differences between our representation learning mechanism and previous typical practices like Zintgraf et al. (2021) are: (1) the target task representation in our approach is a convex combination of the source task representations while that in previous typical practices are non-confined vectors; (2) the task representation for each task is fixed in MATTAR while in previous methods like Zintgraf et al. (2021) the representations are dynamic variables and are not fixed in even one episode. Regarding the advantages of our approach, on one hand, it is more interpretable as the learned linear coefficients offer an intuitive understanding; on the other hand, our practice is more stable to some extent as our representations are defined in a well-formed space instead of being non-confined and the task representation for each task is fixed. To empirically compare our approach and non-confined learning, we adapted Zingtgraf et al. (2021) to our problem setting by combining it with our population-invariant value network and encoder network. The experiment was conducted in *sz* series of the SMAC benchmark and the results are listed as follows where the baseline is denoted as VariBAD. We conduct this small experiment with ten random seeds as suggested by Reviewer dhhQ. Empirically our approach overall outperforms the baseline when transferring to unseen tasks.
>
>
> |               | Source Task 1          | Source Task 2          | Source Task 3          | Unseen Task 1          | Unseen Task 2          | Unseen Task 3          | Unseen Task 4          | Unseen Task 5          |
> | ------------- | ---------------------- | ---------------------- | ---------------------- | ---------------------- | ---------------------- | ---------------------- | ---------------------- | ---------------------- |
> |         | 2s3z                   | 3s5z                   | 3s5z\_3s6z             | 1s8z                   | 1s9z                   | 2s8z                   | 2s9z                   | 7s3z                   |
> | MATTAR  | 0.99$\pm$0.01          | **0.99**$\pm$0.02 | 0.44$\pm$0.18          | **0.80**$\pm$0.08 | **0.65**$\pm$0.15 | **0.90**$\pm$0.09 | **0.87**$\pm$0.06 | 0.16$\pm$0.12          |
> | VariBAD | **1.00**$\pm$0.00 | 0.98$\pm$0.02          | **0.46**$\pm$0.18 | 0.41$\pm$0.29          | 0.26$\pm$0.18          | 0.80$\pm$0.08          | 0.66$\pm$0.22          | **0.20**$\pm$0.11 |

---

### Decision · Program_Chairs · 2023-01-20

**Decision:**

Reject

**Justification For Why Not Higher Score:**

There are some concerns regarding novelty, positioning, and the completeness of the experimental results.

**Justification For Why Not Lower Score:**

N/A

**Metareview: Summary, Strengths And Weaknesses:**

This paper considers the problem of transfer learning/policy reuse across related multiagent problems. This is done by incorporating a latent space that is shared across tasks. The proposed approach is interesting and the clarity of the paper is appreciated, and the reviewers were generally positive regarding this work. On the other hand, they also brought up a few concerns regarding the positioning with respect to earlier work on the topic and experimental design; these should be addressed further before the work is accepted.